# Molecular mechanisms and hotspots of pH sensing in ASIC1a revealed by computational and functional analysis
Olivier Bignucolo[1,2,4], Ophélie Molton[3,4], Ivan Gautschi[3] & Stephan Kellenberger [3] ✉

Extracellular acidification opens the Na⁺-selective acid-sensing ion channels (ASICs), which are key players in neuronal excitation, learning, pain perception, fear conditioning, and ischemic stroke-induced cell death. Their agonists – protons – can theoretically bind to almost any titratable residue, complicating the study of ASIC's activation mechanism. To identify proton binding sites, we developed an ensemble-based computational approach combining Poisson–Boltzmann electrostatics with multiple short MD simulations to calculate pKa values across ASIC1a structures of closed, open, and desensitized states. This method preserves local ion environments and captures solvation-sensitive conformational variability. Predicted pH-sensing residues were conservatively mutated and functionally tested. Residues whose mutation altered pH sensitivity were further analyzed via a palette of substitutions and mathematical modeling, revealing the key side-chain features governing their role. This computational and experimental strategy identified H73 of the wrist, K211 and E242 of the acidic pocket and E375, E413 and E418 of the palm as pH sensors for activation, and E242, E375 and E413 as pH sensors for steady-state desensitization, and revealed how residues of these clusters contribute to channel function. Our findings offer a comprehensive map of pH sensing in ASIC1a and introduce a robust method for investigating pH-dependent mechanisms in other proteins.

Acid-sensing ion channels (ASICs) are neuronal Na⁺-conducting channels activated by extracellular protons. They are involved in learning, fear conditioning and pain perception, and contribute to neurodegeneration[1–3]. Functional ASICs are made of three subunits and can assemble as homo- or heteromeric channels. Four genes encode eight different ASIC subunits in mammals, ASIC1a and -1b[4–7], ASIC2a and -2b[5,8], ASIC3 (in rodents; ASIC3a, -3b and 3c in humans)[9–11] and ASIC4[12–14]. In the central nervous system, homotrimeric ASIC1a is the most pH-sensitive ASIC[15]. Each ASIC subunit comprises a large extracellular domain, two transmembrane α-helices and two intracellular N- and C-termini. The extracellular part of ASIC subunits contains different domains named for its resemblance to a hand holding a small ball (Fig. 1A)[16]. The palm constitutes the extracellular continuation of the transmembrane α-helices and forms together with the knuckle at its top the inner scaffold of the ectodomain. The thumb and finger are α-helix-containing domains that are oriented towards the periphery of the channel, with the finger positioned above the thumb. Each ASIC trimer contains three acidic pockets (AcPs), cavities formed by residues of the thumb, β-ball and finger of one subunit, and residues of the palm

domain of the adjacent subunit (green in Fig. 1A). The central vestibule is enclosed by the β-sheets of the lower palm domains of the three subunits. The junction between the TM domains and the palm is called the wrist; it encloses the extracellular vestibule[16] (Fig. 1A).

Exposure to extracellular acidic pH opens and, in the continued presence of the acidic pH, then desensitizes ASICs. Upon desensitization, the channels enter a non-conducting state. The simplest kinetic scheme describing ASIC's function is thus described by three states: closed, open and desensitized (Fig. 1B). Desensitization from the closed state, called steady-state desensitization (SSD), can occur without apparent channel opening upon exposure to pH values slightly below the physiological pH[1,4]. ASIC activation and desensitization are initiated by protonation events that likely occur in different parts of the ectodomain, inducing conformational changes[17–21] that lead to pore opening and/or desensitization. Different strategies have been used to identify residues contributing to the pH dependence of ASICs. Coric et al. identified gating regions by generating chimeras derived from proton-sensitive and proton-insensitive ASIC1 channels[22], Smith et al. mutated the titratable residues present in proton-

¹Swiss Institute of Bioinformatics, Basel, Switzerland. ²miri dynamics, Basel, Switzerland. ³Department of Biomedical Sciences, University of Lausanne, Lausanne, Switzerland. ⁴These authors contributed equally: Olivier Bignucolo, Ophélie Molton. ✉e-mail: stephan.kellenberger@unil.ch

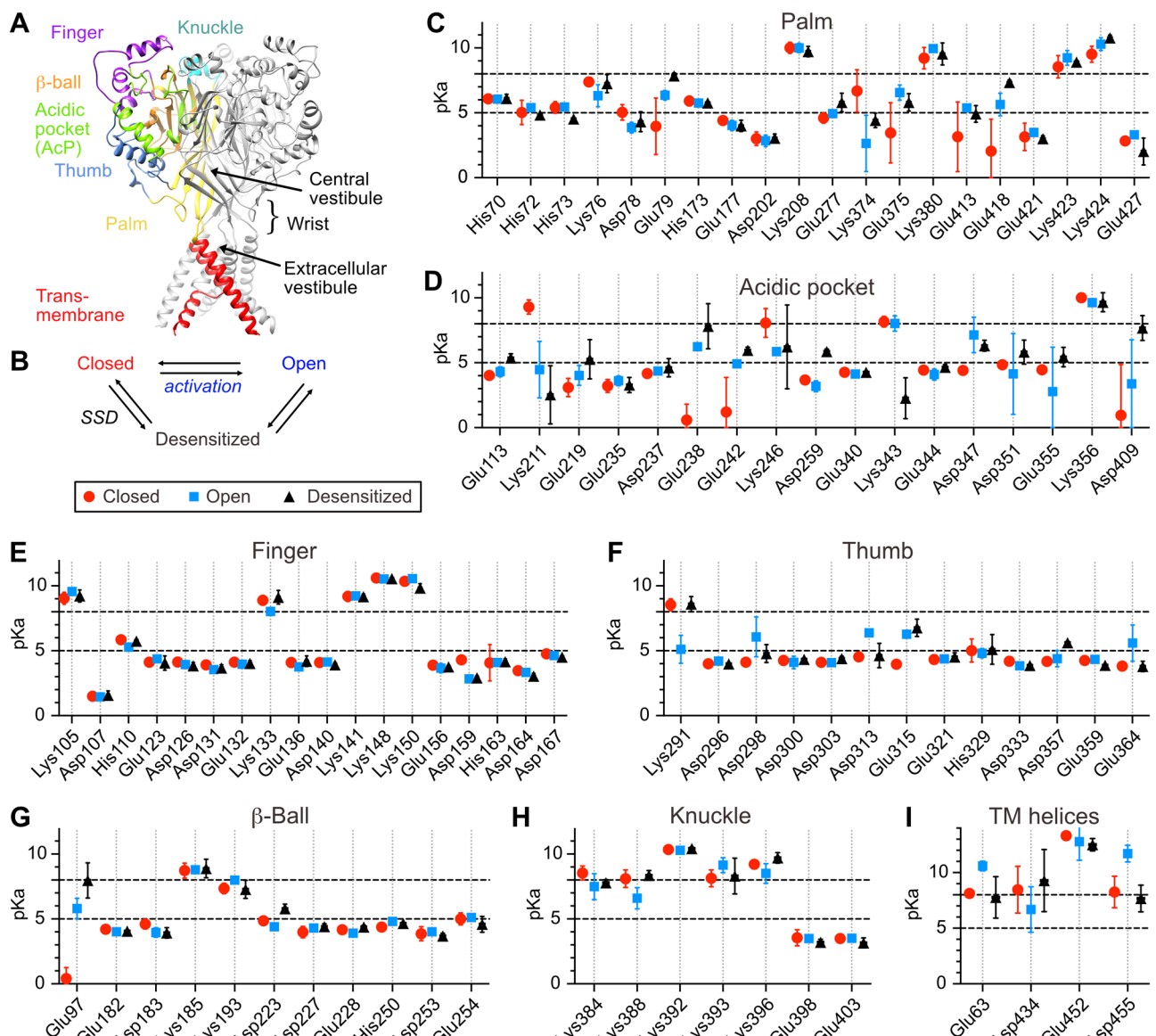

**Fig. 1 | pKa of titratable residues of human ASIC1a. A** Structural image of human ASIC1a (Structural model based on PDB code 5WKU) with the thumb shown in blue, the β-ball in orange, the finger in purple, the knuckle in cyan, the transmembrane domains in red, and the acidic pocket (AcP) in green. **B** The minimal kinetic scheme able to describe ASIC function accurately. The transitions "activation" and steady-state desensitization (SSD) are indicated. **C–I** The calculated pKas (mean ± SD, *n* = 15; see *Methods* for details) of hASIC1a titratable residues are shown for the different regions as presented in the text. Individual symbols correspond to the functional state, with closed state in red, open state in blue and desensitized state in black. Black horizontal dashed lines highlight the pH range of interest.

sensitive ASIC2a but not in proton-insensitive ASIC2b[23], Paukert et al. focused on conserved residues[17], Liechti et al.[20] used pKa calculations based on the desensitized ASIC1a structure available at that time, and Lynagh et al. followed evolution[19]. These studies identified key residues defining ASIC1a pH dependence in the acidic pocket[17,18,20,21,24], the lower palm and the wrist[17,18,20,25] (Table 1 and Supplementary Table 1). The combined substitution of the wrist residues H72 and H73 by Ala suppressed activation[17]. Two studies showed that after simultaneous mutation of 6[18] or 16[21] titratable residues in the acidic pocket of ASIC1a, the channels were still opened by acidification, emphasizing the important role of pH sensors outside the acidic pocket. Residues whose mutation affects the pH dependence may constitute H+ sensors or may affect steps following protonation. In most studies the available data did not allow distinguishing between these two roles.

The aim of the present study was to provide a comprehensive analysis of pH sensing for activation and SSD in ASIC1a. Ensemble-based pKa calculations from structural models of the closed, open and desensitized

conformations of ASIC1a were used to predict potential pH-sensing residues. pKa calculations are often performed uniquely on an experimental structure[20], not considering the conformational changes occurring upon proton binding or unbinding. Furthermore, changes in protonation states of a given residue also affect its electrostatic environment and may therefore alter the likelihood of a neighboring residue to bind or release a proton. These issues are recognized, and there has been an effort to solve them implicitly through rearrangement of charges as a function of the dielectric constants of the protein and the solvent. Yet, these strategies were shown to fail when facing complicated and integral proteins[26,27]. We recently introduced a way to mitigate these problems by designing a strategy involving several successive all-atom short MD simulations[28]. In this approach, protonation states are updated after each round based on the results of electrostatic calculations performed on representative frames. Since these frames are extracted from MD simulations in explicit solvent and the ions located nearby the protein are included in the calculation (Methods), this approach considers solvation and mobile ion effects. The updated

**Table 1 | Comparison with pH50 values of previous studies**

| Residue | This study | | Interpretation other studies | Paukert[a] | | Kellenberger lab | | Other studies | |
|---|---|---|---|---|---|---|---|---|---|
| | Activation | SSD | | Mut | ΔpH50 | Mut | ΔpH50 | Mut | ΔpH50 |
| Glu63 | | | Eff. | Q** | 0.10 | C-AF488[e] | 0.40 | | |
| His70 | | | | | | C-AF488[e] | 0.26 | | |
| His72 | | | Eff. | A | 0.03 | C-AF488[e] | 0.45 | | |
| His73 | Conf. | | Eff. | A | −0.68 | | | C[i]; A[o](ASIC2a) | −1.33; LOF |
| Lys76 | | Put. | Eff. | | | C**[h] | −0.04 | C[p] | −1.44 |
| Asp78 | Eff. | Put. | Eff. | N** | −0.33 | C**[h]; C-AF488[c] | −0.23, −0.21 | A[o](ASIC2a) | LOF |
| Glu79 | Eff. | Eff. | Eff. | Q | −0.11 | C**[h], A[g] | −0.14; −0.72 | Q**[k]; Q[i]; R[o](ASIC2a) | −0.3; −0.8; LOF |
| Glu97 | | Eff. | Eff. | | | Q[d]; A[g] | 0.16; 0.1 | | |
| His110 | | | Eff. | | | C-AF488[c] | 0.04 | A[o](ASIC2a) | LOF |
| His173 | | Put. | | | | | | A[p] | +0.07 |
| Lys193 | | | ND | | | | | | |
| Lys211 | Conf. | | Eff. | M** | −1 | | | M[p], E[i] | −0.8, −0.77 |
| Glu219 | Put. | | Eff. | Q | −0.1 | A[g] | −0.35 | K[k] | 0.05 |
| Glu238 | Eff. | Put. | Eff. | Q | 0 | A[g] | −0.33 | Q[p], K**[k] | −0.2, 0.1 |
| Glu242 | Conf. | Conf. | Eff. | Q | −0.12 | A[g] | −0.34 | | |
| Lys246 | Eff. | Eff. | Eff. | M | −0.16 | | | | |
| Glu254 | | Put. | | Q | −0.07 | Q**[d] | 0.07 | | |
| Asp259 | | Put. | | | | | | A[n] | −0.21 |
| Glu277 | Eff. | Eff. | | | | Q[d] | −0.16 | | |
| Lys291 | | | ND | | | | | | |
| Asp298 | | | | N | 0.08 | | | | |
| Glu315 | Put. | Eff. | Eff. | Q | 0.12 | Q**[d] | −0.06 | | |
| His329 | Put. | Put. | Eff. | N | −0.48 | | | | |
| Lys343 | | | | M | 0.01 | | | AzF[j] | −0.25 |
| Asp347 | Eff. | Eff. | Eff. | | | N**[d], A[g] | −0.19; −0.78 | N**[k], N[m], N[i], N[p] | 0.12, −0.3, −0.29, −0.4 |
| Asp351 | Put. | Put. | Eff. | | | A[g] | −0.32 | N[k], N[m], N[i]; AzF[j] | 0.28, 0.04, −0.17, −0.46 |
| Glu355 | | | Eff. | | | Q**[d], C-AF488[ce], A[g], Q[f] | 0.08; 0.4, 0.3; −0.18; 0.04 | AzF[j] | −0.25 |
| Asp357 | Eff. | Eff. | Eff. | N** | −0.01 | | | AzF[j] | −1.02 |
| Glu364 | | | ND | | | | | | |
| Lys374 | Eff. | Eff. | Eff. | M | −0.23 | | | AzF[j] | −0.5 |
| Glu375 | Conf. | Conf. | Eff. | Q | −0.22 | A[g] | −0.22 | AzF[j] | |
| Lys384 | | Put. | Eff. | M | −0.3 | | | | |
| Lys388 | | Put. | Eff. | M | −0.46 | C-AF488[ce] | 0.18, 0.32 | | |
| Asp409 | Eff. | Eff. | Eff. | N | 0.21 | A[g] | −0.18 | K[k] | 0.17 |
| Glu413 | Conf. | Conf. | | N | −0.09 | Q**[d], A[g] | −0.1; −0.24 | | |
| Glu418 | Conf. | Put. | Eff. | Q | −0.13 | Q**[d], A[g] | −0.2; −0.69 | K[k], Q[i] | −0.15; −0.81 |
| Asp434 | | | Eff. | C[b] | −0.23 | | | | |
| Asp455 | | Put. | Eff. | | | | | AzF[j] | −0.6 |

ΔpH50 = pH50(Mutant)-pH50(WT); "This study": Put. putative pH sensor; Eff. substantial effect (= "functionally important residue"); Conf. confirmed pH sensor; (Eff. and Conf., according to criteria summarized in the discussion). Other columns; **, mutation to several residues tested; ND, not determined; C-AF488, Cys mutation, modified with fluorophore AF488; AzF, crosslinker[78]; LOF, loss of function (in ASIC2a); the upper case single letters indicate residues (single letter code) by which the WT residue was replaced. In "Other studies, "Eff." indicates that the pH50 shift was ≥ 0.15 of a conservative mutation, or ≥0.3 of a non-conservative mutation in ASIC1a.

References: a[17],; b[79],; c[48],; d[20],; e[51],; f[47],; g[43],; h[80],; i[33],; j[78],; k[18],; l[81],; m[16],; n[46],; o[23],; p[19],

protonation pattern is then applied to the next simulation round. This process is repeated until the predicted protonation states become consistent across iterations, that is, when further updates no longer result in changes to the system's charge distribution or local environment. This approach occupies a methodological space between single-frame implicit-solvent electrostatic models and more computationally demanding methods, such as explicit-solvent constant-pH MD, rigorous alchemical free-energy methods, or QM/MM approaches.

After an initial functional screening of predicted pH sensors, residues whose conservative mutation affected the pH dependence (termed here "putative pH sensors") were mutated to different residues, to determine which side chain properties affected the pH dependence. Based on this

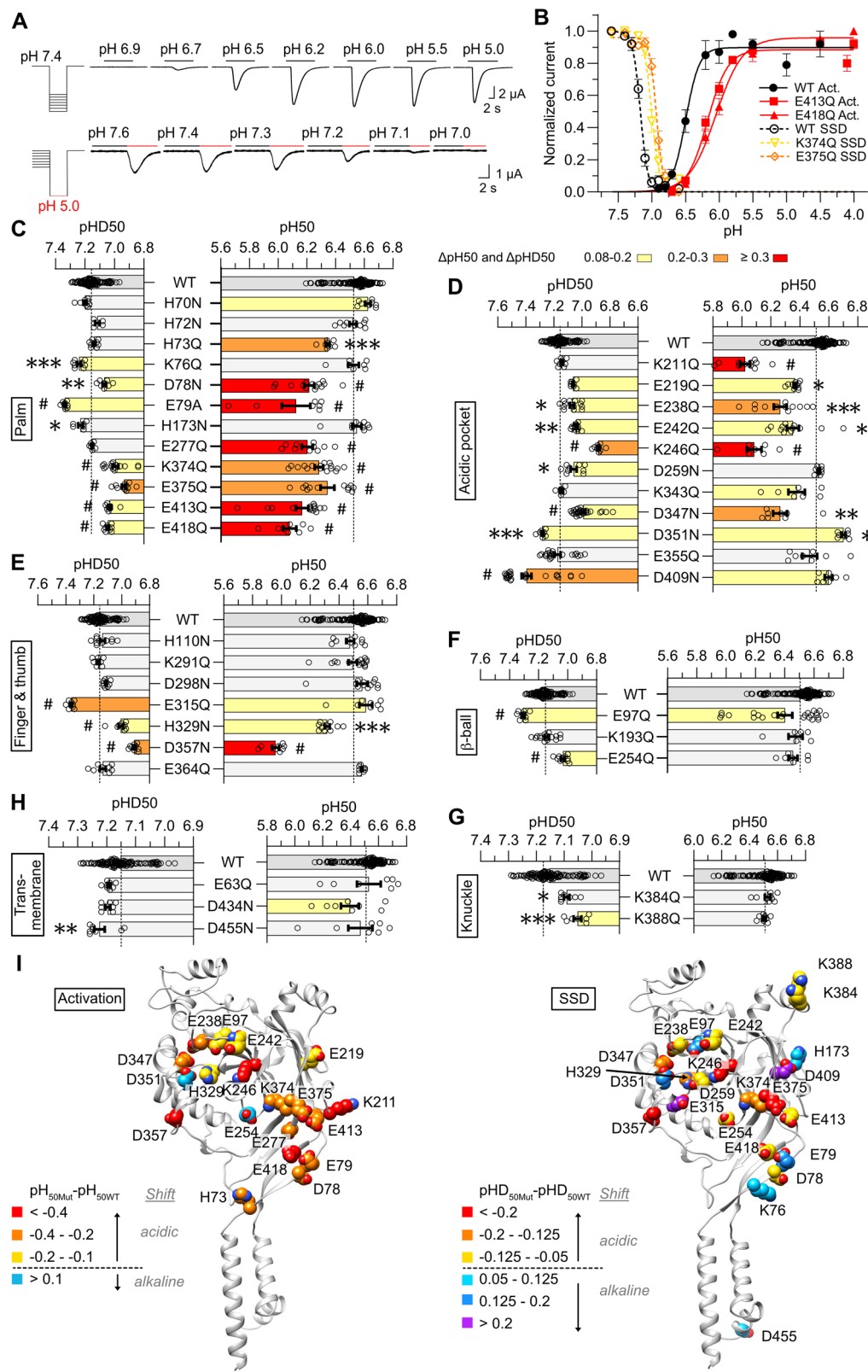

analysis it was concluded whether a given residue was likely a pH sensor or not. Our study identifies novel and confirms previously known residues co-determining the pH ASIC1a dependence, it identifies pH-sensing residues, highlights hotspots of pH sensing in the ASIC ectodomain, and reveals intra- and intersubunit pH sensor pairs that contribute to the activation and desensitization of ASIC1a.

## Results

### pKa-based identification of pH sensors

With the aim of capturing protein dynamics and solvation-sensitive conformational variability, we performed short MD simulations based on the open, closed and desensitized states and calculated the pKa values of titratable residues of human ASIC1a (hASIC1a) on several extracted frames, by

**Fig. 2 | Conservative mutations of 27 predicted pH sensors affect the ASIC1a pH dependence. A** Representative current traces of *Xenopus laevis* oocytes expressing ASIC1a WT, obtained by two-electrode voltage clamp to −60 mV to determine the pH dependence of activation (upper panel) and steady-state desensitization (SSD; lower panel). Channels were activated by a 10-s perfusion with stimulation solution of varying pH every 60-s. The conditioning solution was pH 7.4 between stimulations. The pH dependence of SSD was measured by applying conditioning solutions of varying pH for 50-s before the 10-s stimulation by pH5.0. **B** pH dependence curves of activation (filled symbols) and SSD (open symbols). The WT data (black) were obtained on the same days as the mutants, shown in color. Currents were normalized to the maximum peak current. The lines represent fits to the Hill equation (*n* = 7–15). **C–H** pH values for half-maximal activation ($pH_{50}$) and half-maximal desensitization ($pHD_{50}$) for WT and conservative mutations to non-titratable

residues, obtained by fitting to the Hill equation, in the palm (**C**), AcP (**D**), finger and thumb (**E**), β-ball (**F**), knuckle (**G**) and transmembrane domains (**H**). Comparison of the mutants to WT was done by one-way ANOVA test followed by a Dunnett's multiple comparisons test when > 75% of the groups showed a normal distribution and Kruskall-Wallis test followed by a Dunn's multiple comparison test when ≥ 25% of the groups showed a non-normal distribution; *$p < 0.05$; **$p < 0.01$; ***$p < 0.001$; #$p < 0.0001$. Data are presented as Individual points and as mean ± SEM. The dotted lines represent the WT mean $pH_{50}$ and $pHD_{50}$ values. (**I**) Structure images of the human ASIC1a structural model (based on the open ASIC1a structure 4NTW), presenting residues whose conservative mutation to a non-titratable amino acid significantly shifted the $pH_{50}$ (left panel) or $pHD_{50}$ values (right panel). The color code highlights the amplitude of the shift, as indicated, with yellow to red for acidic, and blue to purple for alkaline shifts.

an ensemble-averaged pKa analysis (see *Methods*). The CHARMM PBEQ solver was used, which determines the electrostatic potential around the biomolecule by solving the Poisson–Boltzmann (PB) equation (see *Methods*). The PB equation combines the Poisson equation, which describes the electrostatic potential in a space containing a charge distribution, with the probability distribution of particles in a system at thermal equilibrium, or Boltzmann distribution. Using the positions of the charges and the partial charges of the biomolecule given by the structure, the electrostatic potential is calculated, which allows modeling the ion distribution in the form of non-uniform ion densities. Thus, the PB equation provides an approach for analyzing the electrostatic interactions between biomolecules and their environment. When the PB equation is solved, it yields the spatial distribution of the electrostatic potential. The resulting electrostatic energies are incorporated into a thermodynamic cycle to estimate the free energy change associated with (de)protonation, from which pKa values are derived. This approach identified residues whose protonation state is modulated by the acidification that triggers ASIC1a activation and SSD. Variations in ion concentration and distribution affect this potential, which in turn determines the protonation states of the amino acids. The PB equation offers a robust framework for linking a protein's electrostatic environment to its protonation behavior (For the derivation of the PB equation and for methodological details, see *Methods*).

The pKas of all Asp, Glu, His and Lys residues of hASIC1a in each of the three conformational states were calculated, and are plotted in Fig. 1, classified by domain. The AcP, to which the finger, thumb and β-ball of one subunit and the palm of an adjacent subunit contribute (Green in Fig. 1A), was initially proposed to be the H⁺-sensor for ASIC activation[16]. We treat here the residues of the AcP as a group, and, for the sake of simplicity, we do not discuss them within their respective domains (i.e. Finger, thumb, β-ball, palm). pKa values were calculated to identify 1) the residues with a pKa within the pH range of either activation or SSD and 2) residues likely to change their protonation state during these transitions. Considering the two transitions activation (Closed→Open, C→O) and SSD (Closed→Desensitized, C→D; Fig. 1B), a protonation change of a given titratable residue can occur at the beginning of the transition due to the pH change (from pH7.4 to typically pH6 for activation and pH6.8 for SSD) when the channel is still in the closed conformation. The conformational transitions to the open or desensitized state change the electrostatic interactions and with this the pKa values of some residues (Fig. 1) and accordingly, the protonation state can further change during these transitions. The probability that a titratable residue is a pH sensor, based on its pKa values, was estimated based on several criteria for the activation and the SSD transition. The application of these criteria generated a scalable output, resulting in a high value in the case of predicted pH sensors. The following factors were considered, detailed here for activation: 1) Relevant pKa range. A change in the protonation state of a titratable residue during activation is likely to occur if the pKa values obtained from the closed and the open structure are within the range of pH5-pH8, or, in case they are outside this range, one value is < pH5 and the other is > pH8. Since the changes in pKa result from altered electrostatic interactions, their occurrence indicates that the conformation in

the proximity of the concerned residue changes during the transition. 2) ΔpKa ( = pKa(O)-pKa(C)) amplitude. A greater change in pKa during the transition suggests a more substantial conformational change. Therefore, the absolute, scaled ΔpKa was considered. Based on these criteria, a ranking was established as illustrated in Supplementary Fig. 1. The pKa of the highest-ranked residue, E238, is 0.58, 6.24 and 7.81 in the closed, open and desensitized conformation, respectively (Fig. 1). Thus, it is deprotonated in the closed state and protonated once the channel has adopted the open or desensitized conformation.

## Measurement of the ASIC1a pH dependence of activation and SSD
Based on the computational prediction of pH-sensing residues (Fig. 1 and Supplementary Fig. 1), 38 titratable amino acid residues were tested for their contribution to activation and SSD. These residues were mutated to non-titratable amino acids of a similar size but no net charge, i.e. Asp and His were mutated to Asn, and Glu and Lys were mutated to Gln. If a given residue is part of a H⁺-binding site involved in activation, preventing its titration by pH should affect the $pH_{50}$ or alter the steepness of the pH dependence of activation. Wild-type (WT) or mutant ASIC1a channels were expressed in *Xenopus laevis* oocytes for functional measurements by two-electrode voltage-clamp. The pH dependence of activation was measured by exposing the channels to stimulation solutions for 10 s, once per minute, applying a series of solutions with increasingly acidic pH (Fig. 2A, upper panel). The conditioning pH between stimulations was 7.4 in all experiments unless noted. Based on such experiments, pH dependence curves of activation were generated (Fig. 2B), indicating an acidic shift with the two mutations E413Q and E418Q (red) compared to WT (black). The pH values of half-maximal activation, $pH_{50}$, were calculated from the curve fit and are plotted in the right half of Fig. 2C. SSD is determined as the remaining channel availability for activation after conditioning at a given pH. The pH dependence of SSD was measured by exposing the channels for 50 s to a conditioning solution before a 10-s stimulation at 5.0 and by repeating this cycle with conditioning solutions of increasingly acidic pH (Fig. 2A, lower panel). Figure 2B compares the SSD of the two mutants K374Q and E375Q (yellow and orange) to that of WT (black, dotted lines), showing an acidic shift in the two mutants, and the corresponding lower $pHD_{50}$ values (Fig. 2C). Some of the mutations induced a sustained current component, shown as an increased sustained current / peak current ratio compared to WT (Supplementary Fig. 2).

## Testing conservative mutations confirms the functional relevance of 27 predicted pH sensors
The screening results of the 38 highest-ranking predicted pH sensors are grouped by domain in Fig. 2, and pH50 and pHD50 values of all mutants are presented in Supplementary Table 2. The impact of these conservative mutations on pH dependence is visually highlighted in Fig. 2C and in subsequent figures by a scale of three colors ranging from light yellow over orange to red (|ΔpH50| ≥ 0.08 and < 0.2, ≥ 0.2 and <0. 3, or ≥ 0.3, respectively), with red indicating the strongest shift. Mutation-induced

**Table 2 | Correlation of pH$_{50}$ and pHD$_{50}$ values with side chain properties**

| | Residue | Number of mutations studied | Activation | | SSD | |
|---|---|---|---|---|---|---|
| | | | positive correlation | Negative correlation | positive correlation | Negative correlation |
| Palm | H73 | 7 | h_do (0.046) | | pol (0.020), h_ac (0.026), coil (0.024) | hyd (0.032), bulk (0.038) |
| | D78 | 7 | pol (0.028), h_ac (0.004) | | | h_do (0.011) |
| | H173 | 4 | | | | |
| | E277 | 4 | | | | |
| | K374 | 7 | size (0.013), surf (0.005), h_do (0.029) | | | |
| | E375 | 5 | | h_do (0.013) | h_ac (0.005) | h_do (0.08) |
| | E413 | 5 | h_ac (0.012) | beta (0.01), h_do (0.015) | h_ac (0.001) | |
| | E418 | 4 | pol (0.052) | | | coil (0.017) |
| Acidic pocket | K211 | 6 | | | | flex (0.011) |
| | E219 | 4 | | | | |
| | E242 | 4 | | | h_ac (0.016) | |
| | K246 | 7 | | | | h_ac (0.016) |
| Thumb | H329 | 7 | | | | |
| Knuckle | K384 | 4 | coil (0.031) | alpha (0.02), bulk (0.026) | | |
| | K388 | 4 | | | size (0.045) | |

Correlations between different scales of amino acid side chain biophysical properties were determined as described in *Methods* and in Supplementary Data 1. The *p* value is indicated in parenthesis. Alpha, α helix propensity; beta, β-sheet propensity; bulk, bulkiness; coil, coil structure propensity; flex, flexibility; h_ac, H$^+$ acceptor, h_do, H$^+$ donor; hyd, hydropathy; pol, polarity; surf, surface.

shifts were generally smaller for SSD than for activation. The results of mutations in the palm are shown in Fig. 2C. Since the mutants E79Q and H73N were not functional, the mutants E79A and H73Q were measured instead. Conservative mutations to a non-titratable amino acids of residues D78, K374, E375, E413 and E418 shifted both the pH$_{50}$ and pHD$_{50}$ toward more acidic values as compared to the WT counterpart, thus decreasing the apparent pH sensitivity (Fig. 2C). The mutation E79A affected the pH dependence of activation and SSD in opposite directions, with an acidic shift in pH$_{50}$ and an alkaline shift in pHD$_{50}$. The mutations H73Q and E277Q shifted the activation pH dependence in the acidic direction and did not affect SSD. Inversely, K76Q and H173N induced an alkaline shift of the SSD without affecting activation.

The strongest pH$_{50}$ shifts of conservative mutations to non-titratable residues in the AcP were observed with K211Q and K246Q (Fig. 2D). Conservative mutations of E238, E242, K246 and D347 induced acidic shifts in both pH$_{50}$ and pHD$_{50}$, while mutating D351 resulted in alkaline shifts of both pH$_{50}$ and pHD$_{50}$. Several mutations affected either activation or SSD. K211Q and E219Q induced an acidic shift of the pH$_{50}$. The D259N mutation induced an acidic, and D409N an alkaline shift of the pHD$_{50}$. Conservative mutations to non-titratable residues of the thumb residues H329 and D357 induced acidic shifts of pH$_{50}$ and pHD$_{50}$, while the E315Q mutation shifted the pHD$_{50}$ to more alkaline values without affecting activation pH dependence (Fig. 2E). Of the tested mutants in the β-ball, the pHD$_{50}$ was shifted towards alkaline values by the E97Q mutation and towards acidic values by the E254Q mutation (Fig. 2F). Conservative mutations of the knuckle residues K384 and K388 induced small acidic shifts of the pHD$_{50}$ (Fig. 2G). In the transmembrane segments, the D455N mutant showed a small alkaline shift in pHD$_{50}$ (Fig. 2H).

The functional screening analysis identified out of the 38 predicted pH sensors a total of 27 residues whose mutation affected the pH dependence. We name them "putative pH sensors". Of these, 4 residues are involved in activation only, 10 in SSD only, and 13 in both. Aligning these mutations as a function of their pH$_{50}$ showed acidic shifts with many mutants, with the

most negative ΔpH$_{50}$ of −0.55, while only few mutations induced an alkaline shift in pH$_{50}$ (most positive ΔpH$_{50}$ of +0.19; Supplementary Fig. 3A). Strongest negative shifts were observed with D357N (thumb), E418Q (palm), K211Q, K246Q (both AcP), followed by several palm mutations, including E413Q, E277Q, D78N, H73A, K374Q and E79A. ΔpHD$_{50}$ values of conservative mutations ranged from −0.27 to +0.24 (Supplementary Fig. 3B). The acidic shifts with the highest amplitude were measured with K246Q (AcP), D357N (thumb), E375 (palm) and D347N (AcP), while the most alkaline shifts were observed with D409N (AcP), E315Q (thumb), E79A (palm) and E97Q (β-ball). To show where in the structure mutations induced acidic and alkaline shifts, Fig. 2I uses a color code to indicate in a structural image of one subunit the amplitude of pH$_{50}$ shifts (of activation; left panel) and of pHD$_{50}$ shifts (of SSD; right panel). This shows that acidic shifts of the pH$_{50}$ are most frequent in palm mutations, while negative and positive shifts of the SSD pHD$_{50}$ are more equally distributed between domains.

**Approach for correlating functional data with residue side chain properties and general framework for data interpretation**

Many identified putative pH sensors were mutated to several different amino acid residues to determine which of their side chain properties are most important for their functional role. For residues for which ≥ 4 substitutions had been generated, we tested whether the shift in pH dependence correlated with 12 scales of side chain properties: bulkiness, flexibility, hydrogen acceptor, hydrogen donor, hydropathy, polarity, probability of occurring in α helices, probability of occurring in β sheets, probability of occurring in β turns, probability of occurring in a coil, size, and surface (*Methods* and Table 2). This analysis allows to link the pH dependence to the side chain properties at a given position, opening the way to hypotheses and mechanisms beyond charge modification.

In the interpretation of the functional effects of these mutations, we include the calculated protonation fraction (f(prot), 1=fully protonated, 0=deprotonated) of the original side chain at the start and end of a given

**Table 3 | Protonation fraction of predicted pH sensors**

| Domain | State/transition | F(protonated) | | | | |
|---|---|---|---|---|---|---|
| | | Closed<br>C | Start activation<br>C→O | End activation<br>O | Start SSD<br>C→D | End SSD<br>D |
| | Considered pH | 7.4 | 6 | 6 | 6.8 | 6.8 |
| | Structure from which calculated | Closed | Closed | Open | Closed | Desensitized |
| Palm | His70 | 0.05 | 0.55 | 0.53 | 0.16 | 0.17 |
| | His72 | 0.00 | 0.10 | 0.20 | 0.02 | 0.01 |
| | His73 | 0.01 | 0.20 | 0.22 | 0.04 | 0.01 |
| | Lys76 | 0.49 | 0.96 | 0.67 | 0.79 | 0.74 |
| | Asp78 | 0.00 | 0.10 | 0.01 | 0.02 | 0.00 |
| | Glu79 | 0.00 | 0.01 | 0.69 | 0.00 | 0.92 |
| | His173 | 0.03 | 0.45 | 0.36 | 0.11 | 0.08 |
| | Glu277 | 0.00 | 0.04 | 0.08 | 0.01 | 0.09 |
| | Lys291 | 0.93 | 1.00 | 0.11 | 0.98 | 0.98 |
| | Lys374 | 0.16 | 0.83 | 0.00 | 0.44 | 0.00 |
| | Glu375 | 0.00 | 0.00 | 0.78 | 0.00 | 0.02 |
| | Glu413 | 0.00 | 0.00 | 0.19 | 0.00 | 0.01 |
| | Glu418 | 0.00 | 0.00 | 0.30 | 0.00 | 0.77 |
| Acidic pocket | Lys211 | 0.99 | 1.00 | 0.03 | 1.00 | 0.00 |
| | Glu219 | 0.00 | 0.00 | 0.01 | 0.00 | 0.03 |
| | Glu238 | 0.00 | 0.00 | 0.63 | 0.00 | 0.91 |
| | Glu242 | 0.00 | 0.00 | 0.08 | 0.00 | 0.13 |
| | Lys246 | 0.82 | 0.99 | 0.42 | 0.95 | 0.21 |
| | Lys343 | 0.85 | 0.99 | 0.99 | 0.96 | 0.00 |
| | Asp347 | 0.00 | 0.03 | 0.93 | 0.00 | 0.25 |
| | Asp351 | 0.00 | 0.07 | 0.01 | 0.01 | 0.10 |
| | Glu355 | 0.00 | 0.03 | 0.00 | 0.00 | 0.04 |
| | Asp409 | 0.00 | 0.00 | 0.00 | 0.00 | 0.88 |
| Finger | His110 | 0.03 | 0.41 | 0.16 | 0.10 | 0.08 |
| Thumb | Asp298 | 0.00 | 0.01 | 0.54 | 0.00 | 0.01 |
| | Glu315 | 0.00 | 0.01 | 0.65 | 0.00 | 0.48 |
| | His329 | 0.00 | 0.09 | 0.06 | 0.02 | 0.02 |
| | Asp357 | 0.00 | 0.01 | 0.02 | 0.00 | 0.06 |
| | Glu364 | 0.00 | 0.01 | 0.28 | 0.00 | 0.00 |
| β-ball | Glu97 | 0.00 | 0.00 | 0.31 | 0.00 | 0.94 |
| | Lys193 | 0.47 | 0.96 | 0.99 | 0.78 | 0.75 |
| | Glu254 | 0.00 | 0.09 | 0.12 | 0.02 | 0.01 |
| | Asp259 | 0.00 | 0.00 | 0.00 | 0.00 | 0.10 |
| Knuckle | Lys384 | 0.93 | 1.00 | 0.97 | 0.98 | 0.91 |
| | Lys388 | 0.83 | 0.99 | 0.80 | 0.95 | 0.97 |
| Trans-membrane | Glu63 | 0.84 | 0.99 | 1.00 | 0.95 | 0.90 |
| | Asp434 | 0.92 | 1.00 | 0.83 | 0.98 | 1.00 |
| | Asp455 | 0.88 | 0.99 | 1.00 | 0.97 | 0.88 |

The protonation fraction was calculated for the 5 conditions indicated, based on the pKa values determined in the closed, open and desensitized structural models for the pH conditions indicated in the table. The Henderson-Hasselbalch equation was used to calculate the protonation fraction from the pH and from the pKa of the residue in a given ASIC conformation.

transition, based on the Henderson-Hasselbalch equation (*Methods*, Table 3). For activation for example, we consider f(prot) in the closed state (pH 7.4, pKa calculated from the closed structural model), at the start of the opening transition (pH 6.0; pKa calculated from the closed structural model) and at the end of the transition when the channel is in the open conformation (pH 6.0, pKa calculated from the open conformation structural model). Five conditions are considered in total for activation and SSD, as shown in Table 3.

## Titratable residues of the wrist shape the pH dependence of activation

The location of predicted pH sensors in the wrist, at the interface of two subunits, is shown in Fig. 3A. The functional measurements confirmed the relevance for the pH dependence of H73, K76 and D78 (Fig. 2C), identifying them as "putative pH sensors" (bold labels in Fig. 3A). The relatively low pKa values of the wrist residue H73 (Fig. 1C, Table 3) suggest that it is neutral in the closed and desensitized conformations and partially

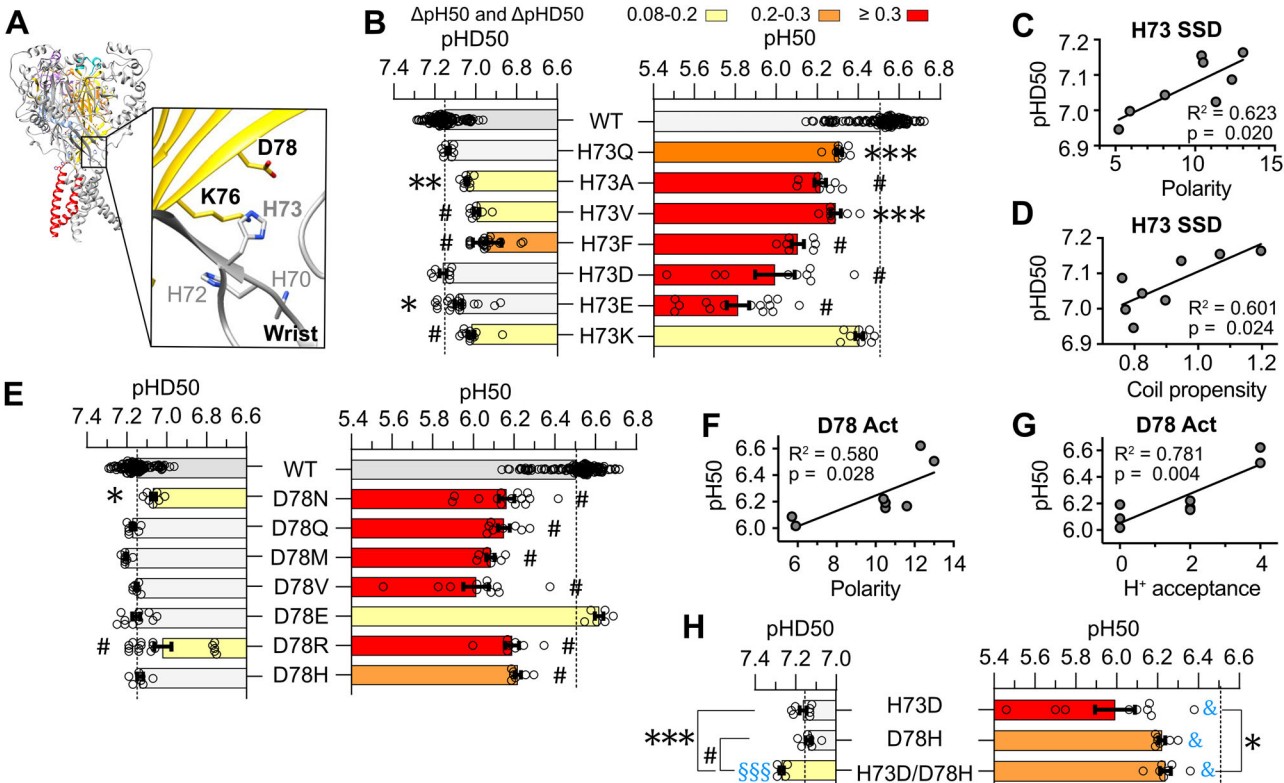

**Fig. 3 | Role of side chain properties of putative pH sensors of the wrist.**
**A** Structural images of the wrist domain in the closed conformation of ASIC1a (based on PDB 5WKU), showing the location of the predicted pH sensors. Of these, the putative pH sensors ( = predicted pH sensor whose conservative mutation to a non-titratable amino acid affected the pH dependence) are labeled in bold. The different domains are indicated by specific colors in one of the three ASIC subunits (see Fig. 1A). **B** $pH_{50}$ and $pHD_{50}$ values of WT ASIC1a and indicated mutants of H73. **C, D** Dependence of the $pHD_{50}$ on side chain properties polarity (**C**) and coil propensity (**D**) at position H73. **E** $pH_{50}$ and $pHD_{50}$ values of WT ASIC1a and

indicated mutants of D78. Dependence of the $pH_{50}$ on the side chain properties polarity (**F**) and $H^+$ acceptance (**G**) at position D78. **H** $pH_{50}$ and $pHD_{50}$ values of the acidic-basic residue swap mutant H73D/D78H and the corresponding individual mutants. For the $pH_{50}$ and $pHD_{50}$ data, the same statistical tests as in Fig. 2C were carried out; *$p < 0.05$; **$p < 0.01$; ***$p < 0.001$; #$p < 0.0001$. The dotted lines represent the WT mean $pH_{50}$ and $pHD_{50}$ values. For the analysis of acidic-basic residue swap mutants, the difference to WT is indicated with blue symbols as §$p < 0.5$; §§$p < 0.01$; §§§$p < 0.001$; &$p < 0.0001$. Data are presented as Individual points and as mean ± SEM.

protonated upon activation (f(prot) ≈ 0.20). The acidic shifts of $pH_{50}$ and $pHD_{50}$ were indistinguishable between the mutations of H73 to hydrophobic residues of different sizes (Fig. 3B). Mutation of H73 to acidic or basic residues induced an acidic shift of the $pH_{50}$, except for H73K which retains titratability, and a small acidic or no shift of the $pHD_{50}$, indicating no clear role for the charge of the residue. The correlation analysis with biophysical properties (*see Methods*) of the seven H73 mutants indicates that activation may be favored when a hydrogen bond donor occupies this position (Table 2), and that the desensitized state is favored with hydrophilic or polar residues, hydrogen bond acceptors and residues with high coiled coil propensity at position 73 (Fig. 3C, D, Table 2). D78 is located close to H73 of the neighboring subunit. Due to its low pKa it is negatively charged, thus unprotonated, in the three conformations (Fig. 1C, Table 3). Similarly to H73, most mutations of D78, except mutation to Glu, induced substantial, similar acidic shifts of the $pH_{50}$, including mutations to hydrophobic or hydrophilic residues ($\Delta pH_{50} \geq 0.3$ pH units, Fig. 3E). Mutational effects on SSD were very small. The $pH_{50}$ correlated positively with polarity and $H^+$ acceptance (Fig. 3F, G, Table 2). The strong effects of mutating D78 to Arg or His, but not to Glu, indicates that the most important property of position 78 for activation is the presence of a carboxylic acid functional group.

The minimal distance between the side chain heavy atoms of H73 and D78 of the neighboring subunit in the closed, open and desensitized states is 4.2, 5.5 and 3.1 Å (Fig. 3A, Supplementary Table 3), respectively. This suggests an interaction between the two residues via a salt bridge, or by an $H^+$-bond, as seen in the desensitized chicken ASIC1a structure[16]. The double mutant H73C/D78C was reported to have a reduced pH sensitivity for

activation, while incubation of H73C/D78C with the reducing agent dithiothreitol doubled the current amplitudes, suggesting the requirement of varying interactions between these two residues for efficient channel opening[19]. The acidic-basic residue swap mutant H73D/D78H had a similar $pH_{50}$ as the single mutant D78H and rescued partially the acidic shift of the H73D mutation (Fig. 3H). According to the principle of mutant cycle analysis, an interaction between two residues is expected if the effect of the double mutation is different from the sum of the effects of the two corresponding single mutations[29,30]. Although this approach has its limitations[31], it is a powerful test of interactions. The non-additive effects of these two mutations support an interaction between H73 and D78 during activation. The two individual mutations did not change the $pHD_{50}$, unlike the acidic-basic residue swap mutation which induced an alkaline $pHD_{50}$ shift of 0.12 ($p < 0.001$), supporting therefore also an interaction in the closed-desensitized transition, consistent with the shortest distances observed in the desensitized state (Supplementary Table 3).

### Residues of the central and lower palm co-determine the pH dependence of activation and SSD

K374 is located on the palm β-strand 10, near the channel's central vertical axis, at ~30 Å above the plasma membrane. It faces the central channel axis, E375 of the β-strand 10 and E413 located in a loop structure between β-strands 11 and 12, both of a neighboring subunit (Fig. 4A). All tested K374 mutations shifted the activation and SSD pH dependence to more acidic values, except for K374R for activation and K374M for SSD (Fig. 4B). The $pH_{50}$ shifts correlated strongly with the size ($p = 0.013$, $R^2 = 0.67$) and the

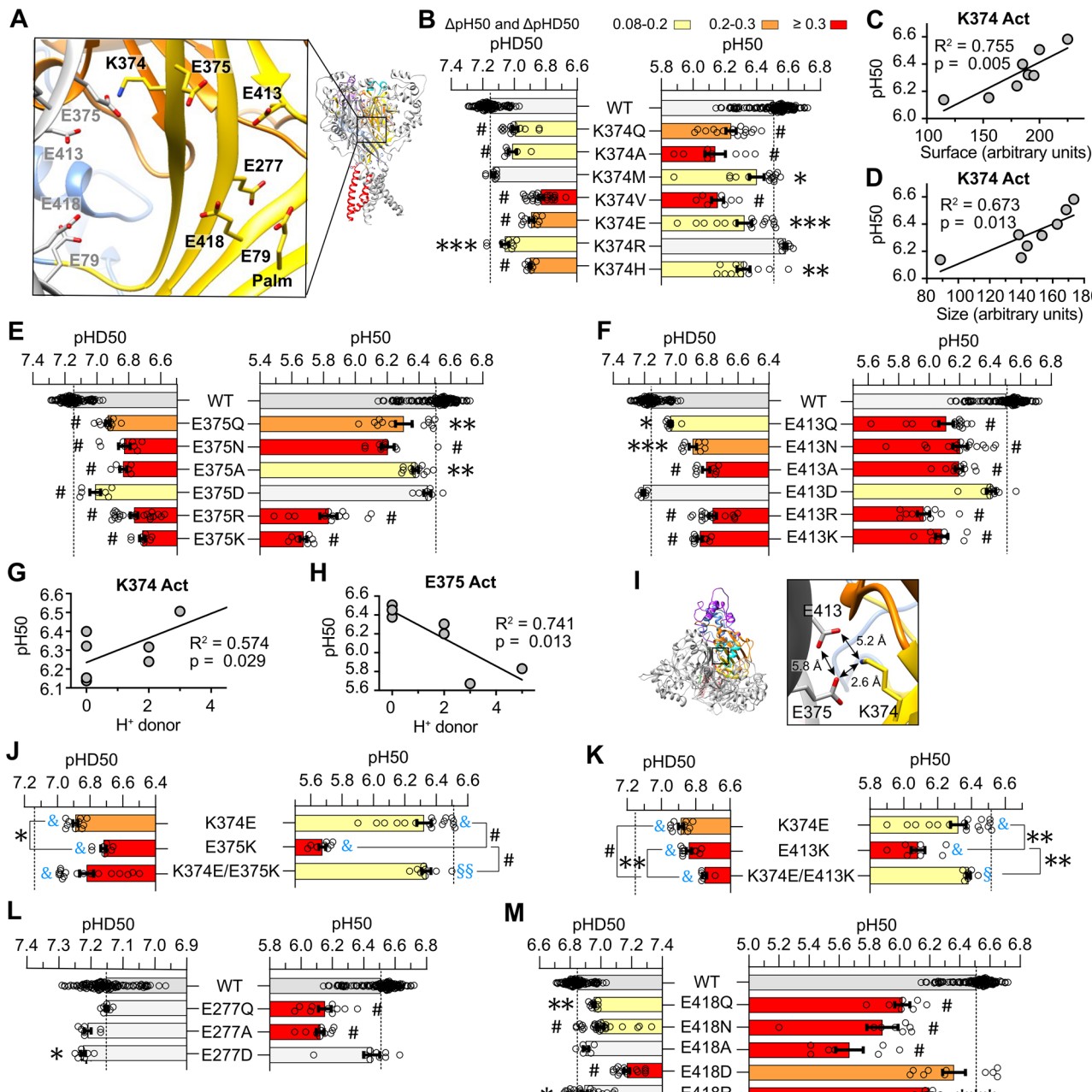

**Fig. 4 | Role of side chain properties of putative pH sensors in the central palm. A** Structural image of the ASIC1a palm domain in the closed conformation (based on 5WKU), showing the location of the predicted pH sensors, with the putative pH sensors labeled in bold. **B, E, F, L, M** $pH_{50}$ and $pHD_{50}$ values of WT ASIC1a and mutants of residue K374 (**B**), E375 (**E**), E413 (**F**), E277 (**L**) and E418 (**M**). **C, D, G** Dependence of the $pH_{50}$ on the side chain surface (**C**), size (**D**) and "H⁺ donor" property (**G**) at position 374. **H** Dependence of the $pH_{50}$ on the side chain property H⁺ donor at position 375. **I** Detailed view of residues K374, E375 and E413, with indication of distances between closest side chain heavy atoms (based on

5WKU). **J, K** $pH_{50}$ and $pHD_{50}$ values of the acidic-basic residue swap mutant K374E/E375K and of the corresponding single mutants (**J**) and of K374E/E413K and of the corresponding single mutants (**K**). For the $pH_{50}$ and $pHD_{50}$ data, the same statistical test as in Fig. 2C were carried out; *$p < 0.05$; **$p < 0.01$; ***$p < 0.001$; #$p < 0.0001$. The dotted lines represent the WT mean $pH_{50}$ and $pHD_{50}$ values. For the analysis of acidic-basic residue swap mutants, the difference to WT is indicated with blue symbols as §$p < 0.5$; §§$p < 0.01$; §§§$p < 0.001$; &$p < 0.0001$. Data are presented as Individual points and as mean ± SEM.

surface area ($p = 0.005$, $R^2 = 0.76$), indicating that a large residue is required at this position (Fig. 4C-D, Table 2). As with K374, most mutations of E375 and E413 shifted the $pH_{50}$ and $pHD_{50}$ to more acidic values (Fig. 4E-F). Mutation to Asp had no or only a small effect on activation and SSD, indicating that the side chain length is not critical at these two positions. On E375, the strongest acidic pH dependence shifts were measured when Glu was replaced by a basic residue, while on E413 in contrast, such acidic-basic inversions induced similar shifts as did mutations to Ala or Asn. Low pH increases K374 protonation in the closed state, potentially driving activation

and SSD, yet K374 appears to be neutral in the open and desensitized state (Table 3). The regression analyses based on the functional studies show that this apparent contradiction is likely due to proton sharing occurring upon activation. They suggest that the open state is stabilized when position 374 is occupied by a H⁺ donor and destabilized when a H⁺ donor is located at position 375 (Fig. 4G-H, Table 2). The K374R mutation did not affect the $pH_{50}$, with Arg being always positively charged, indicating that titratability of this residue is not essential for normal ASIC1a activation. The pKa calculations indicate that E375 is protonated in the open state. Among the

E375 mutations tested, E375D was the only one that did not shift the $pH_{50}$, highlighting the importance of an acidic residue at this position. The K374 and E375 residues are close enough for H-bonding (Supplementary Table 3, Fig. 4I). In further support of a tight interaction between K374 and E375 residues of neighboring subunits, the acidic-basic residue swap mutant K374E/E375K largely rescued the acidic shift in $pH_{50}$ of mutation E375K, showing a similar $pH_{50}$ as the single mutant K374E and decreasing the $\Delta pH_{50}$ relative to WT from $-0.83$ (E375K, $n = 8$) to $-0.17$ (K374E/E375K, $n = 8$; Fig. 4J). We conclude that one of the three protons of the Lys374 amino group moves closer to Glu375 upon activation. E413 is located at the edge of a β strand. The $pH_{50}$ values of E413 mutants negatively correlated with β-strand propensity (Table 2). A similar partial suppression of the acidic shift in $pH_{50}$ of E413K was found with the acidic-basic residue swap mutant K374E/E413K, which reduced the $\Delta pH_{50}$ relative to WT from $-0.42$ to $-0.12$ (Fig. 4K). An $H^+$ acceptor is important at position 413 (Table 2). Although these two double mutants did not fully restore WT properties, they largely rescued the $pH_{50}$ shift of the single charge inversion mutations of E375 and E413. We conclude that K374 shares a proton with both E375 and E413 upon activation, in a case of a triple interaction. The three residues are close enough to form hydrogen bonds (Supplementary Table 3 and Fig. 4), making several protonation mechanisms plausible. One possibility is a dynamic equilibrium, where the proton remains largely associated with a specific donor (e.g., K374) but rapidly shifts hydrogen bonding between the two glutamate residues (E375/E413) over time. Alternatively, delocalized protonation may occur, where the proton is not localized to a single residue but spreads across the hydrogen bond network, as in aromatic rings. A third possibility is direct hydrogen bonding without proton transfer, where strong static hydrogen bonds form between residues but the proton remains localized.

In the context of SSD, both E375 and E413 show a positive correlation between $pHD_{50}$ and $H^+$ acceptance (Table 2). The acidic-basic residue swap mutation K374E/E375K showed a small, partial recovery of the acidic $pHD_{50}$ shift by E375K, while K374E/E413K showed an approximately additive effect of the two individual mutations (Fig. 4K), suggesting a probable interaction of between K374 and E375 and no interaction between K374 and E413 during SSD. These observations are consistent with the shorter distance of K374 to E375 than to E413, and the further increase of the K374-E413 distance upon desensitization (Supplementary Table 3).

E277 is located on the β-strand 9 of the lower palm, below E413 (Fig. 4A) and is negatively charged in the three conformations. Mutation to Ala and Gln induced an acidic shift of ~0.3 pH units on activation, whereas the E277D mutation did not change activation and induced a small alkaline shift in SSD pH dependence (Fig. 4L). This suggests a role for the titratability or of the negative charge for activation. E79 and E418 are located further down in the lower palm (Fig. 4A). The E79A and E418Q mutations shifted the $pH_{50}$ to more acidic values, while for $pHD_{50}$ it induced an alkaline shift in E79A and an acidic shift in E418Q (Fig. 2C). According to the pKa values, both residues are protonated only once the channel adopts the open or desensitized conformation. Since these residues have been previously investigated in detail in many studies[18,20], we tested only a small selection of E418 mutations (Fig. 4M). In the context of activation, the E418A mutation to hydrophobic residues induced a stronger shift than mutation to the positively charged Arg. The $pH_{50}$ values showed a tendency towards a positive correlation with polarity ($p = 0.052$, $r^2 = 0.653$; Table 2, Fig. 4M). Shifts of the SSD pH dependence were small, except for a $\Delta pHD_{50}$ of $-0.33$ of the E418D mutation. The small amplitude of pHD50 shifts prevented a detailed interpretation. A previous study had suggested, based on the analysis of 11 different side chains at position 418, that large and hydrophilic residues favored the closed, while small and hydrophobic residues favored the desensitized state[20].

**Titratable acidic pocket residues contribute to activation and SSD**

In the AcP, which collapses upon activation, many acidic residues are concentrated in a small space. All predicted pH sensors of the AcP (Fig. 5A)

except for K343 and E355 contributed to ASIC1a pH dependence (Fig. 2D), with conservative mutations to non-titratable amino acids of K211, K246 and D347 showing the strongest effects. K211 points from the central palm of one subunit towards the thumb of the adjacent subunit (Fig. 5A). Based on its pKa (Fig. 1D and Table 3), K211 is deprotonated only in the open and desensitized states. All generated K211 mutants, including K211R, induced a strong, similar acidic shift of the $pH_{50}$ (Fig. 5B). Contrary to K374, the comparison of $pH_{50}$ values with side chain properties at position 211 did not yield any correlations, suggesting that a precise geometry of the side chain is required or that titration is essential. K211 is close to a predicted $Cl^-$ binding site in the neighboring subunit[16], which introduces additional electrostatic interactions. K211 was previously shown in rat ASIC1a to bridge the palm and thumb domains of adjacent subunits by forming a hydrogen bond between its side chain ε amine and the backbone carbonyl oxygen of L351 (corresponding to L353 in hASIC1a)[19]. MD simulations have previously shown that K211 may form a salt bridge with the backbone atoms of D357 of the thumb-palm loop at the lower, inner end of α helix 5 of the adjacent subunit, highlighting an intersubunit interaction[32]. In the structural models of hASIC1a, the closest K211-D357 distance decreases from ~7 Å in the closed to ~4 Å in the open state. In the closed state, K211 is within < 6 Å of the thumb residues E315 and D351 of the adjacent subunit. E315 is located on the α4 helix, and D351 of the α5 helix is oriented towards the vestibule of the AcP (Fig. 5A). Upon activation, the K211-E315 distance increases slightly, while the K211-D351 distance increases to 11.3 Å due to the collapse of the AcP. E315 is negatively charged in the closed conformation at both physiological and acidic pH, and partially protonated in the open and desensitized states, while D351 is always negatively charged (Fig. 1D, E and Table 3). This suggests attractive electrostatic interactions between K211 and these two acidic residues in the closed state and at the onset of opening or desensitization. In the open and desensitized states in contrast, the proton between these two residues is closer to D351, resulting in neutral D351 and K211 side chains. Acidic-basic inversion mutations of these three residues induced a strong acidic $pH_{50}$ shift with E315K ($-1.28$, $n = 9$) and smaller acidic shifts with K211E ($-0.47$, $n = 9$) and D351K ($-0.37$, $n = 11$) (Fig. 5C, D). In both E315K and D351K, the double acidic-basic inversion mutation with K211D did not rescue the effect of the single mutation on activation, suggesting that the residues do not interact. The effects of these mutations on SSD were small, indicating that electrostatic interactions of K211, E315 and D351 are less important for SSD. The SSD experiments showed however a partial rescue by the double mutations, indicating a possible interaction during this transition (Fig. 5C, D).

E242 and K246 are in the inner upper corner of the AcP, in proximity of E219 and D409 of a neighboring subunit. K246 is positively and E242 and D409 are negatively charged in the closed state at pH7.4 and immediately after acidification (Fig. 1D, Table 3), suggesting an attraction of K246 to E242 and D409. In the open and desensitized states, K246 is deprotonated but D409 is protonated, resulting in neutral K246 and D409 side chains. Mutation of K246 to Ala, Met, Val and Glu lowered both the $pH_{50}$ and $pHD_{50}$ while mutation to Arg showed no effect (Fig. 5E), indicating that a positive charge at position 246, but not titratability, is important for activation. The comparison with side chain property scales indicated a negative correlation for the $pHD_{50}$ value with $H^+$ acceptor potential (Table 2). Diverse substitutions induced substantial acidic shifts of $pH_{50}$ and $pHD_{50}$ if done at position 242, and smaller effects at position 409 (Fig. 5F–H). The small effects of D409 mutations suggest that titratability is not required at this position. In contrast, the similarly large acidic shifts of E242 mutations to Gln, Ala, Asp and Lys suggest a role for E242 titratability and side chain geometry. K246 may form a salt bridge with E242 and/or D409, to which the closest side chain heavy atom distances in the closed state are ~6.8 Å (Supplementary Table 3). The K246-D409 distance remains similar, while the K246-E242 distance increases in the open and desensitized conformation. The acidic-basic residue swap mutant K246E/E242K exhibited an essentially additive effect of the single mutations in the context of both activation and SSD (Fig. 5I). In contrast, the mutant K246D/D409K exhibited the same acidic $pH_{50}$ shift as the single mutation K246E for

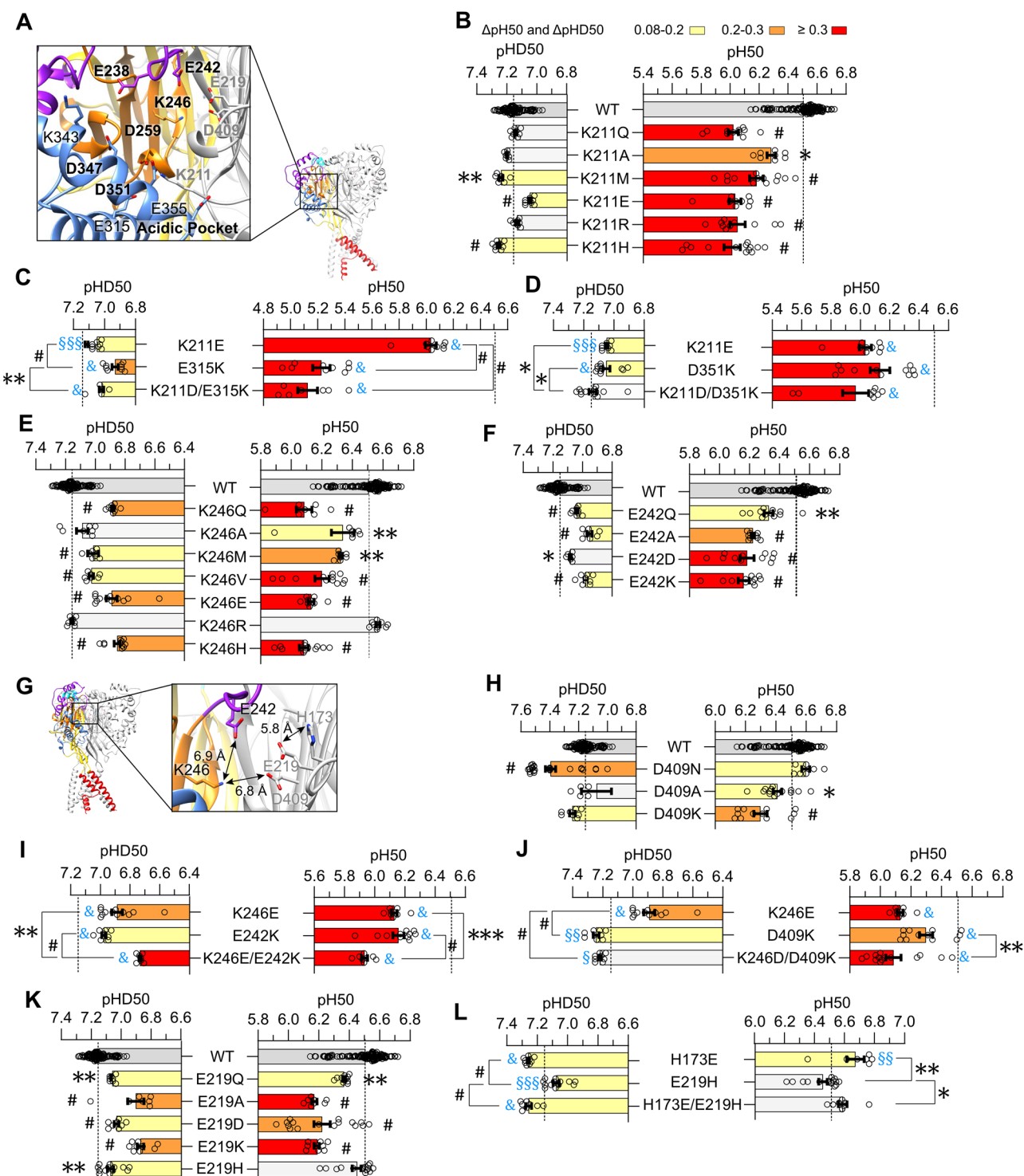

**Fig. 5 | Analysis of putative pH sensors in the acidic pocket. A** Structural images of the AcP of ASIC1a in the closed conformation (based on 5WKU), showing the location of the predicted pH sensors, with the putative pH sensors labeled in bold. **B–F** $pH_{50}$ and $pHD_{50}$ values of WT ASIC1a and mutants of residue K211 (**B**), the acidic-basic residue swap mutant K211D/E315K and of the single mutants K211E and E315K (**C**), the acidic-basic residue swap mutant K211D/D351K and of the single mutants K211E and D351K (**D**), of mutants of K246 (**E**) and E242 (**F**). **G** Structural image showing the location of E219, E242, K246, D409 and H173, based on the structural model of closed ASIC1a (5WKU). **H–L** $pH_{50}$ and $pHD_{50}$ values of WT ASIC1a and mutants of D409 (**H**), the acidic-basic residue swap mutants K246D/E242K (**I**), K246E/D409K (**J**), of mutant E219 (**K**) and the acidic-basic residue swap mutants H173E/E219H (**L**) together with the corresponding single mutants. For the $pH_{50}$ and $pHD_{50}$ data, the same statistical tests as in Fig. 2C were carried out; *$p < 0.05$; **$p < 0.01$; ***$p < 0.001$; #$p < 0.0001$. The dotted lines represent the WT mean $pH_{50}$ and $pHD_{50}$ values. For the analysis of acidic-basic residue swap mutants, the difference to WT is indicated with blue symbols as §$p < 0.5$; §§$p < 0.01$; §§§$p < 0.001$; &$p < 0.0001$. Data are presented as Individual points and as mean ± SEM.

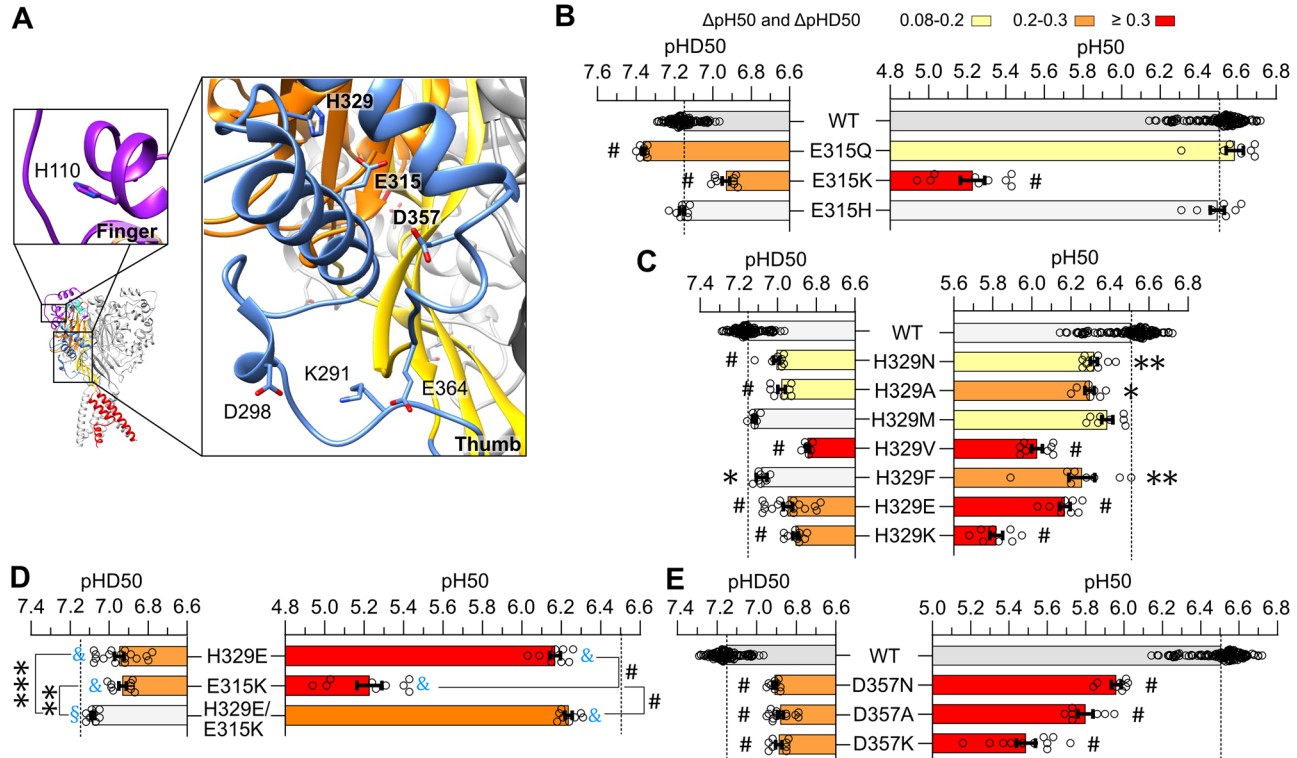

**Fig. 6 | Analysis of putative pH sensors in the thumb and finger domains. A** The location of predicted pH sensors in the finger and thumb domains of ASIC1a are shown in the closed conformation (based on 5WKU), showing the location of the predicted pH sensors, with the putative pH sensors labeled in bold. **B–E** $pH_{50}$ and $pHD_{50}$ values of WT ASIC1a and mutants of residue E315 (**B**), H329 (**C**), the acidic-basic residue swap mutant H329E/E315K and of the corresponding single mutants

(**D**) and mutants of residue D357 (**E**). For the $pH_{50}$ and $pHD_{50}$ data, the same statistical tests as in Fig. 2C were carried out; *$p < 0.05$; **$p < 0.01$; ***$p < 0.001$; #$p < 0.0001$. The dotted lines represent the WT mean $pH_{50}$ and $pHD_{50}$ values. For the analysis of acidic-basic residue swap mutants, the difference to WT is indicated with blue symbols as §$p < 0.5$; §§$p < 0.01$; §§§$p < 0.001$; &$p < 0.0001$. Data are presented as Individual points and as mean ± SEM.

activation but rescued the effect of K246E on SSD (Fig. 5J), suggesting therefore possible interactions between K426 and D409 in the SSD transition.

H173, E219 and D409 are located side by side on the antiparallel β strands 3, 6 and 11 respectively, with similar side chain orientations, and E219 positioned between H173 and D409 (Fig. 5G). Based on the pKa values, E219 is negatively charged in the three conformations, whereas H173 becomes partially positively charged at pH6 (Fig. 1C, D and Table 3), consistent with a weak attraction between H173 and E219 at low pH. The E219 substitutions shifted the $pH_{50}$ and $pHD_{50}$ to more acidic values, except for E219H, which left the $pH_{50}$ unchanged and caused a slight acidic shift in $pHD_{50}$ (Fig. 5K). Mutation to Lys induced a similar $pH_{50}$ shift as the mutation to Asp or Ala, confirming that titratability between a neutral and a negatively charged side chain at position 219 is not critical (Fig. 5K). The regression analyses did not reveal which property of the position 219 side-chain affects the transitions (Table 2). Mutations of H173 induced either an alkaline or no shift of $pH_{50}$ and $pHD_{50}$ values (Supplementary Fig. 4A, B). The distance between the nearest side chain heavy atoms of H173 and E219 is 5.4–5.9 Å in the three conformations (Supplementary Table 3 and Fig. 5G). The $pH_{50}$ of the acidic-basic residue swap mutant H173E/E219H is not different from the WT and corresponds approximately to the sum of the shifts of the individual mutations, suggesting that the two residues do not interact in this transition. The double mutant displayed an alkaline shift of the $pHD_{50}$ that might be due to a dominant effect of H173E (Fig. 5L).

D347 and D351 of the α5 thumb helix are oriented towards a loop containing the acidic residues D237 (not shown) and E238 (Fig. 5A). D351 is always negatively charged, while D347 is partially protonated only in the open and desensitized states. Conservative mutations to non-titratable residues induced alkaline shifts at position 351 and acidic shifts at position 347 (Fig. 2D). The acidic shift in $pH_{50}$ was stronger with Lys than with Glu,

while the $pHD_{50}$ value was shifted in a similar way by the two mutations, indicating that these two positions require an acidic residue for activation, but not for SSD (Supplementary Fig. 4C, D).

## Several thumb residues not pointing into the AcP contribute to the pH dependence

The location of predicted pH sensors in the finger and thumb domains outside the AcP is shown in Fig. 6A. Of the putative pH sensors in these domains, E315 is located at the center of the α4 helix, H329 is part of the α4-α5 linker at the upper, outer end of α helix 4, and D357 is part of the thumb-palm loop at the lower, inner end of α5. H329 is neutral in all conformations, D357 is always negatively charged, while E315 is partially protonated in the open and desensitized states (Fig. 1F and Table 3). Mutation of E315 to Gln and His did not change the $pH_{50}$, whereas mutation to Lys induced a very large acidic shift (Fig. 6B), showing that a basic residue at this position profoundly affects activation, but that the absence of an acidic residue is well tolerated. The E315K mutation induced a smaller acidic shift in $pHD_{50}$, consistent with an impairment of the transition in the presence of a basic residue. Most H329 mutations induced acidic shifts in activation and/or SSD (Fig. 6C), but neither the $pH_{50}$ nor the $pHD_{50}$ values correlated with side chain properties (Table 2). The acidic-basic residue swap mutant H329E/E315K rescued almost completely the acidic shift of the single E315K mutation on activation and of the individual mutations on SSD (Fig. 6D and Supplementary Table 3), supporting the conclusion that these residues, which are 4.6–5.8 Å apart (Supplementary Table 3), interact with each other, likely forming a salt-bridge. Mutations of D357 – which is negatively charged in the three conformations – caused substantial acidic $pH_{50}$ shifts, increasing in the order of Asn, Ala and Lys, and lowering the $pHD_{50}$ by an equal extent by the 3 mutations, establishing together D357 as an important determinant of pH dependence (Fig. 6E). Mutation of the

Finger residue H110 (Fig. 6A) to Lys but not to Asn induced acidic shifts of the $pH_{50}$ and $pHD_{50}$ (Supplementary Fig. 4E).

## Small effects of mutations of predicted pH sensor residues in the β-ball and knuckle

Of the predicted pH sensors in the β-ball (Fig. 1G and Supplementary Fig. 4G), conservative mutations of E97 and E254 affected the pH dependence, causing however only small shifts (Fig. 2F and Supplementary Fig. 4H). All tested mutations of the putative pH sensors K384 and K388 located in the knuckle showed no or only small acidic shifts (Supplementary Fig. 4G and 4I-J). The small effects of mutations of these β ball and knuckle mutations indicate that these residues are of minor importance for the ASIC1a pH dependence. Mutations to non-titratable amino acids of the three predicted pH sensors in the transmembrane ASIC part affected the pH dependence only in the case of D455 (Fig. 2H). Due to its intracellular location, this residue, which contributes to the ion selectivity[33,34] was not further studied.

## Overlap of pH sensing for activation and SSD

A plot of the $\Delta pHD_{50}$ of each tested mutation as a function of its $\Delta pH_{50}$, organized by domain in supplementary Fig. 6, shows that for the domains in which sufficient mutants were tested, the palm, the AcP and the thumb, there is a significant positive correlation between these values, indicating that many residues that shape the pH dependence of activation have a similar role for SSD.

## Discussion

We used here an ensemble-based computational approach combining Poisson–Boltzmann electrostatics with multiple short MD simulations to estimate pKas on structural models of ASIC1a in its three functional states to rank titratable residues by their likelihood of acting as pH sensors. In-depth mutational and functional analysis of the higher ranked residues identified residues H73, K211, E242, E375, E413 and E418 as pH sensors of activation, and E242, E375 and E413 as pH sensors for SSD, with several other titratable residues interacting with these sensors and contributing to pH-dependent gating.

This study combined rounds of short MD simulations and calculations using the Poisson-Boltzmann equation solver of CHARMM to calculate the pKa of structural models of hASIC1a in its three main functional states. We allowed the models of ASIC1a extracted from crystal structures in their three main functional states to relax at RT and in the presence of solvent, but without letting them undergo major conformational changes. To this aim we developed a new strategy that leverages the advanced capabilities of the PBEQ solver to inform our selection of protonation states, while incorporating a series of successive short MD simulations and a progressive adjustment of protonation states until no change of protonation state is required, as explained in the *Methods*. Similarly to previous work which addressed the conformational flexibility with MD simulations[35,36], our approach was chosen because it prioritizes optimal accuracy within computational limits. We have shown previously[28] that in our hands this combination was more powerful at identifying experimentally known key residues than other tools like PropKa[37] and H + +[38].

So far, most studies investigating pH dependence have focused on Asp, Glu and His residues as potential pH sensors. Circular dichroism and optical spectroscopy have demonstrated that Lys residues, whose pKa in water is 10.4, can exhibit significantly altered pKa values if buried within proteins[39,40]. Such large pKa shifts were not observed for Arg and Tyr residues. We had included Tyr in the pKa calculations from the three states and Arg in the calculations based on the closed and desensitized states but did not measure any pKa value close to the gating range of ASICs. Consequently, only the pKa values of all Asp, Glu, His and Lys residues in the protein were further considered. We show here that three Lys residues contribute importantly to the ASIC1a pH dependence, K211 and K246 of the AcP and K374 of the palm. K211, was shown to be a confirmed pH sensor. Since mutation to Arg, which is generally never deprotonated, did not affect the pH dependence

when introduced at residues K246 and K374, we conclude that titratability is not required at these two positions.

Mutation of many residues, including those not directly involved in proton binding, may affect the pH dependence of ASIC currents. A mutation that changes the pH sensitivity may do so by deleting a protonation site, or, if the mutated residue is not a pH sensor, by changing local electrostatic interactions and protein conformation around a pH sensor. It may also affect conformational changes that are downstream of the protonation events. We applied the following rational, multi-step approach to identify titratable residues that are pH sensors (Supplementary Fig. 5): 1) pKa values were calculated based on closed, open and desensitized structural models and the potential of being pH sensors was estimated based on the pKa values, considering the pH changes that induce ASIC1a activation and SSD. The best-ranked residues were termed "predicted pH sensors". 2) The pH dependence of mutations of predicted pH sensors to the closest non-titratable amino acid was analyzed experimentally, identifying 27 residues, whose conservative mutation affected the pH dependence of activation and/or SSD, as "putative pH sensors" (Fig. 2I and Supplementary Fig. 5). Putative pH sensors were further subdivided: 3) If the functional analysis of a palette of mutations of a putative pH sensor showed that the titratability of the residue was important for its function, we termed them "confirmed pH sensors". 4) Several putative pH sensors, most of them being in proximity of confirmed pH sensors, are functionally important (we consider here residues as functionally important, if for conservative mutations, $|\Delta pH50| \geq 0.2$ or $|\Delta pHD50| \geq 0.15$ compared to WT) but did not fulfil the criteria for confirmed pH sensors. Some of these residues were not confirmed as pH sensors either because not enough substitutions were studied (e.g. E79, D347) or because the functional results argued that titratability is not an important factor (e.g. K374 and K246, where the mutation to Arg, which cannot be titrated, did not change the pH dependence). Several of the residues not confirmed as pH sensors are intimately linked to confirmed pH sensors and contribute, possibly without changing their protonation status, to ASIC gating.

Figure 7 highlights the key residues in the five pH-sensing hotspots for activation, the AcP, lower thumb, central and lower palm and wrist, summarizing the main parameters per residue. Confirmed pH sensors are framed in red, whereas functionally important putative pH sensors (as defined above) are shown in a black frame. In the AcP, the basic residues K211 and K246 and the acidic residues E238, E242 and D347 co-determined the pH dependence of activation, while in the lower thumb, D357 shapes the pH dependence. In the lower palm, E79, E277, K374, E375, E413 and E418 contribute to activation. For SSD, functionally important putative pH sensors and confirmed pH sensors include residues of the AcP (E242, K246, D347, D409), the lower thumb (E315, D357), β ball (E97), and the central and lower palm sites (E79, K374, E375, E413), while the wrist residues do not contribute substantially (Supplementary Fig. 7). Of these, E242, E375 and E413 are confirmed pH sensors of both activation and SSD, while H73, K211 and E418 are confirmed pH sensors of activation.

Our results with acidic-basic residue swap double mutants emphasize the essential intersubunit interactions of K374 with E375 and E413 in the central palm. As discussed above, these residues are close enough for hydrogen bonding (Supplementary Table 3). In the lower palm, E79 and E418 of the three subunits interact with each other and contribute to the pH dependence, as shown previously[18,20,41]. In the wrist, the confirmed pH sensor H73 of one subunit interacts with D78 of the neighboring residue, to drive activation. The H73-D78 pair forms hydrogen bonds in the desensitized state, whereas the two residues are farther apart in the closed and open states.

The calculation of the protonation fraction f(prot) (Table 3) identifies residues whose f(prot) changes 1) at the moment of acidification, thus possibly driving the transition ("driver") - if their f(prot) of the closed structure changes between pH7.4 and pH6.0 - or 2) once the open or desensitized conformation is reached, thus possibly stabilizing the newly attained state ("stabilizer") - if their f(prot) changes between closed structure at pH7.4 and open structure at pH6.0. This classification is entirely based on

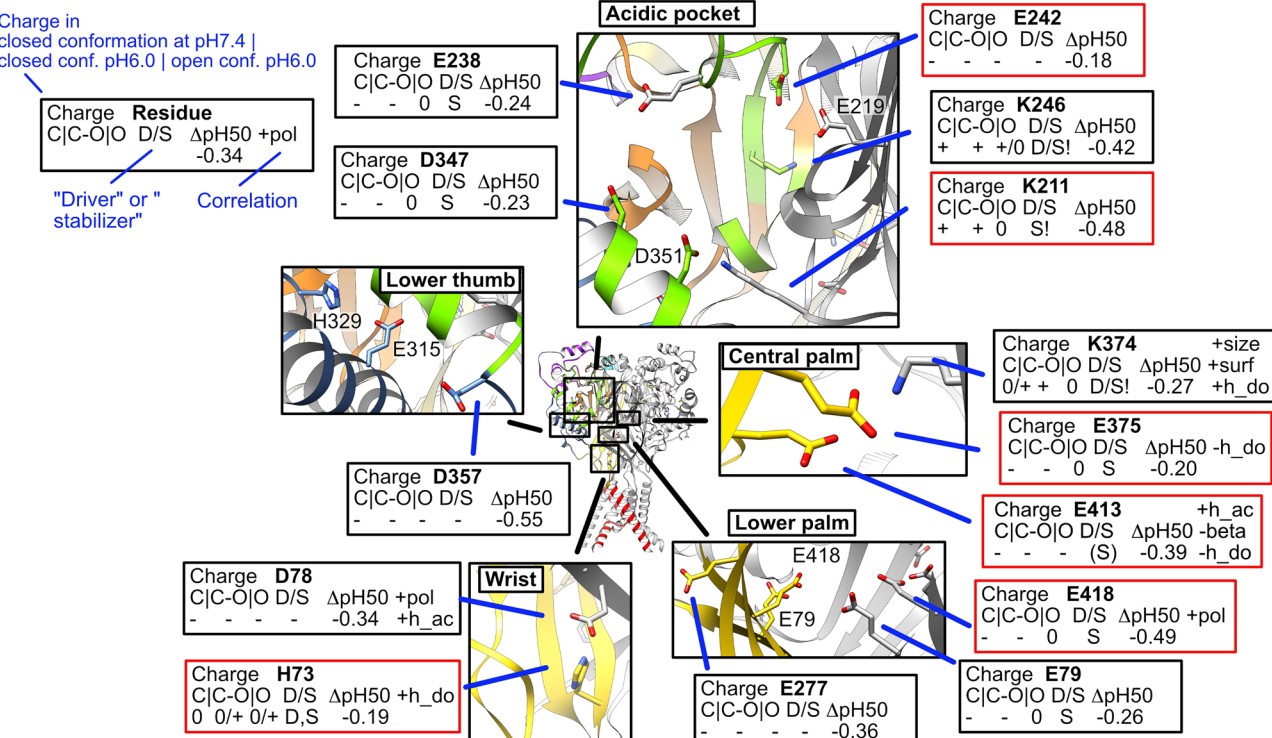

**Fig. 7 | Hotspots of pH sensing for ASIC activation.** Summary of the structural context, computational predictions and functional results of confirmed pH sensors (framed in red) and functionally important putative pH sensors (black frame, defined as follows: for conservative mutations, $|\Delta pH50| \geq 0.2$ or $|\Delta pHD50| \geq 0.15$ compared to WT) but do not fulfil the criteria for confirmed pH sensors, as defined in the text). The five H-sensing regions AcP, lower thumb, central palm, lower palm and wrist are shown in focused structural images, whose locations are indicated in an image of the entire ASIC1a structure (center). As illustrated in the upper left corner, black and red boxes provide the following information on the key residues: Charge, charge of the side chain based on the calculation of the protonation fraction (Table 3) for the situations C = closed structure, pH7.4; C-O, closed conformation, pH6.0 (initiation of transition); O, open structure, pH6.0. D/S, based on the changes in protonation fraction, a residue is expected to drive the transition (D) or stabilize the open state (S). An "!" indicates that acidification induces a deprotonation of this residue. $\Delta pH_{50}$, change in $pH_{50}$ of the conservative mutation to a non-titratable amino acid. Correlations to biophysical side chain properties (Table 2) are indicated with the + sign if the correlation is positive, - if there is a negative correlation; h_ac, hydrogen acceptor; h_do, hydrogen donor; pol, polarity, beta, probability of occurring in β sheets; size, size; surf, surface.

the pKa calculations. Of the putative pH sensors for activation, residues H73, K246 and K374 may drive activation ($|\Delta f(prot)| \geq 0.15$ [taken here as arbitrary limit]) and H73, E79, K211, E238, K246, D347, K374, E375, E413 and E418 may stabilize the activated state, as predicted by the changes in f(prot). The analogous analysis of the C→D transition indicates that of the putative pH sensors for SSD, K76 and K374 likely drive, and K76, E79, E97, E238, K246, E315, D347, K374, D409 and E418 likely stabilize the desensitized state, because their f(prot) increases once the desensitized conformation is reached. It is of interest to note that, for the Lys residues K211, K246 and K374, the f(prot) is lower in the open or desensitized compared to the closed conformation, indicating that the pKa calculations predict that these residues are mostly neutral when the channel is in the open or desensitized conformation. As shown above, these are cases of proton sharing, in which a proton carried by a Lys residue at high pH moves towards one or more interacting acidic residues (K374→E375/E413; K246→E242/D409). For some residues, the changes in protonation fraction are not compatible with their functionally determined role as pH sensors. For the functionally confirmed pH sensors of activation, only H73 shows a protonation change compatible with driving the transition; K211, E375, E413 and E418 change their protonation fraction during the transition, suggesting that they may rather stabilize than drive activation (Table 3 and Fig. 7). While our classification of titratable residues into "functionally important pH sensors" and "confirmed pH sensors" relies essentially on functional criteria, the f(prot) values and distinction into drivers and stabilizers of activation or SSD, are based on the computational results. The

conclusions of the functional and the computational approach are consistent for most, but diverge for some residues.

Most of the residues reported here as putative and confirmed pH sensors were previously shown, in separate studies, to affect pH dependence when mutated, as discussed in the introduction and documented in Table 1. Some of these studies showed the dependence of the pH50 on side chain properties at a given position, complimentary to the information provided here. The modification of D347C by either a negatively or positively charged or uncharged sulfhydryl reagents induced acidic shifts of the $pH_{50}$, and no or small effects on pHD50, underlining the contribution of the titratability, size and polarity of the D347 side chain to ASIC1a activation[20,41]. Studies using mutations, or Cys mutations combined with modification by different types of sulfhydryl reagents concluded that the acidic function and hydrophilicity of E413 and E418 codetermine the pH dependence[18,20].

The novelty of the present study lies in its comprehensive approach that uses a rational strategy for initial selection of predicted pH sensors and the in-depth mutational/functional analysis of such residues. With this analysis, we show that the wrist, palm and AcP are the most important domains involved in pH sensing for activation. The paired acidic-basic residue swap mutations, together with the estimation of residue protonation and mutation to a palette of amino acid residues, highlight interactions at key positions of the channel, such as between K374 and E375 and E413, and between H73 and D78, and provide insights into the pH sensing by interacting basic and acidic residues. The structure-function aspect of the SSD pH dependence has so far almost exclusively been analyzed by our

laboratory. Residues identified here, not previously associated with SSD pH dependence, are E242, K246 and D347 of the AcP, D357 of the lower thumb and K374 and E375 of the palm.

To estimate the risk of having missed in this study any important pH sensor, Supplementary Table 1 lists all extracellular and transmembrane titratable residues not functionally investigated here, together with any published functional data addressing their role. Residues, whose mutation changed the pH50 by ≥0.15 (conservative mutation) or ≥0.3 (any mutation) in publications, and were not studied here, are (in hASIC1a numbering), D107, E123, E136 (finger), E182 and D253 (β-ball), D202 and K380 (palm), E235 and D237 (AcP), D303 and K356 (thumb), K392, K396 and E403 (knuckle) and K423 and E427 (wrist). Several of these residues are in proximity of residues whose functional importance was shown in our study, suggesting that they may interact with pH-sensing hotspots. Other residues are isolated and have a low pKa in the three states, suggesting that they are negatively charged in all pH conditions, as e.g. D107 on the finger, whose mutation strongly shifted the pH50[17]. In such cases, the acidic residue is not titrated during ASIC gating, but its negative charge or the carboxyl group is important for channel function. 16 extracellular titratable residues of ASIC1a have so far not been functionally characterized (Supplementary Table 1). Most of these residues are highly solvent-exposed (Supplementary Table 1) and are probably not titrated in the ASIC activation cycle.

$Ca^{2+}$ has been shown to compete with protons for binding sites in ASICs, stabilizing the closed conformation, as evidenced by an acidic shift of the pH dependence with increasing extracellular $Ca^{2+}$ concentration[42–44]. In ASIC1a, $Ca^{2+}$ binding sites were identified in the AcP, palm and the extracellular pore entry[43]. The following residues identified here as confirmed pH sensors (or functionally important putative pH sensors) are also part of $Ca^{2+}$ binding sites: E242 (E238, D347) of the AcP and E79, E375, E413 and E418 of the palm.

Channel activation and desensitization processes are correlated with rearrangements within subunits and at subunit interfaces. The AcP contains many titratable residues. Their protonation likely leads to the collapse of the AcP, as indicated by structure comparison[16,45] and resonance energy transfer and voltage-clamp fluorometry experiments[21,46–48]. Conformation changes in the AcP can be transduced to the transmembrane domain via the interaction between the β-turn and the upper end of the TM1[16,49]. The Palm domain is involved in conformational changes during activation and desensitization[45,50]. Opening and desensitization involve a displacement of the β-strands 1 and 12 and conformational changes in the structures directly above them. During desensitization, the lower palm β sheets undergo a closing movement, and there are 180° orientation changes of a peptide bond in the β1- β2 linker and of two side chains in the β11-β12 linker just above the β-strands 1 and 12. Despite these large structural changes, the inter-subunit distance between K374 and E375 residues does not change much (Supplementary Table 3). Our extended analysis indicates that K374 interacts with E375 and E413 with intersubunit H-bonds, and that K374 acts as a $H^+$ donor and E375 and E413 as $H^+$ acceptors, suggesting that dynamic protonation/deprotonation occurs during gating transitions. During activation, the β1 and β12 strands of the lower palm domain are displaced to induce opening of the pore. H73, interacting with D78 of the neighboring subunit is at an important position involving these conformational changes. The conformational changes in the wrist are likely driven by protonation events in the AcP and the palm and mediated towards the wrist via the lower palm and the β-turn - TM1 interactions. This is supported by VCF measurements showing evidence of conformational changes with the kinetics of channel opening at distal sites and in the wrist[51]. H73 contributes together with D78 to proton sensing; these residues are near each other in the three states ( ≤ 5.5 Å) and they likely participate in driving the opening of the pore.

## Conclusions
Based on this and previous studies we conclude that titratable residues of the AcP, the palm and the wrist are the key determinants of pH dependence, with additional sparse contributions from finger and lower thumb α helices, the β ball and the knuckle. Residues that we identify here as confirmed pH

sensors are in the AcP, palm and wrist. These are Asp, Glu, His and Lys residues. pH sensing depends on a cooperative contribution of several residues in the 3 domains, including in addition to the confirmed pH sensors also residues whose function as pH sensors was not confirmed in our study, tagged here "functionally important putative pH sensors", and requiring a tight interaction between confirmed pH sensors and such residues. In the palm and wrist, pH sensing occurs at several levels (K374/E375/E413; E79/E277/E418 and H73/D78). It is likely that close to the pore (lower palm and wrist), such residues are involved in conformational changes induced by distal pH sensors, and at the same time influence the transition or the stability of the newly reached conformation by their change in protonation state. Our in-depth analysis of ASIC1a pH sensing adds new elements towards a molecular description of the ASIC activation mechanism, and highlights critical sites within the channel that could serve as targets for new drugs.

## Methods
### Derivation of the Poisson–Boltzmann equation for pKa calculations
Understanding the protonation state of key residues in transmembrane proteins, which is defined by their pKa value and the pH, is crucial for deciphering their functional mechanisms. The pKa depends on several factors, such as nearby charged residues, hydration polarity of the environment. Molecular dynamics (MD) simulations offer atomic-level insights into protein structures and dynamics, but accurately predicting pKa values in the complex protein environment requires considering electrostatic interactions with the surrounding solvent, ions and, in the case at hand, the cell membrane. In this context, previous work has shown that pKa calculations bear the potential to help identify proton sensors in a variety of proteins.

The Poisson equation is a fundamental model used to calculate the electrostatic potential around biomolecules in solutions. It relates the Laplacian of the electrostatic potential ($\varphi$) to the charge distribution ($\rho$), and the dielectric constant ($\varepsilon$) of the medium. They all depend on the location in space, highlighted here by the vectorial notation **r**:

$$\nabla^2 \varphi(\boldsymbol{r}) = -\frac{\rho(\boldsymbol{r})}{\varepsilon(\boldsymbol{r})} \tag{1}$$

In the context of a frame extracted from a MD simulation, the charge density arises from the distribution of charged atoms and molecules in the system. For computational efficiency, the non-protein regions are treated implicitly by the PBEQ module. We made use of this functionality, attributing specific dielectric constants to the solvent and the core and head-group regions of the membrane (see below). It is however possible to introduce explicitly specific entities beyond the protein. As shown in the next section, we used this possibility to explicitly treat the ions located in the vicinity of the protein in MD-extracted frames. The dielectric constant in this equation also depends on the environment, with values differing strongly between the membrane and the solvent. This variability arises because water molecules are polar and have a strong dipole moment, allowing them to easily reorient in response to an electric field, thereby reducing the field strength. Additionally, water molecules form a network of hydrogen bonds, further reducing the effective electric field strength. Conversely, lipid molecules are weakly polarized, and their fatty acid chains do not harbor significant dipole moments, resulting in minimal modification of the electric field.

Solving the PB equation allows us to estimate the electrostatic potential of each atomic position, considering the effect of the surrounding environment.

The Boltzmann distribution relates the probability of occurrence of a charge at a given location in respect to the potential experienced at that location. In this context, it provides the probability of finding ions as a

function of their location around the charged residue:

$$\frac{c(\mathbf{r})}{c_\infty} = e^{\frac{-E(\mathbf{r})}{k_BT}} \tag{2}$$

where:

$c(\mathbf{r})$ is the molar concentration of ions at $\mathbf{r}$ defined by its components $\mathbf{r_x}$, $\mathbf{r_y}$, and $\mathbf{r_z}$,

$c_\infty$ is the molar concentration in the bulk,

$E(\mathbf{r})$ is the deviation from equilibrium energy at the considered location,

$k_BT$ is the Boltzmann scaling factor, where $k_B$ is the Boltzmann constant and $T$ is the temperature.

Considering that $E(\mathbf{r}) = ze\varphi(\mathbf{r})$, we can rewrite the concentration in terms of potential:

$$c_i(\mathbf{r}) = c_{\infty,i}\exp\left(-\frac{z_i e\varphi(\mathbf{r})}{k_BT}\right) \tag{3}$$

where:

$e$ is the elementary charge

$z_i$ is the valence of ion species i

$C_{\infty,i}$ is the bulk concentration of ion species i

Substituting the Boltzmann distribution into the Poisson equation, we obtain the Poisson-Boltzmann equation. The following version is used in the context of solutions containing multiple ion species:

$$\nabla^2\varphi(\mathbf{r}) = -\frac{1}{\varepsilon(\mathbf{r})}\left(\rho(\mathbf{r}) + \sum_i z_i eC_{\infty,i}\exp\left(\frac{-z_i e\varphi(\mathbf{r})}{k_BT}\right)\right) \tag{4}$$

The PBEQ module addresses this n-body problem numerically through finite difference estimations. The $e^{-x}$ expression can be simplified using a Taylor expansion, where it is assumed that the handled potentials are sufficiently small to retain only the first term:

$$e^{-x} \approx 1 - x \tag{5}$$

This allows us to rewrite a simplified form of Eq. (4):

$$\nabla^2\varphi(\mathbf{r}) \approx -\frac{1}{\varepsilon(\mathbf{r})}\left(\rho(\mathbf{r}) + \sum_i z_i eC_{\infty,i}\left(1 - \frac{z_i e\varphi(\mathbf{r})}{k_BT}\right)\right) \tag{6}$$

We multiply both sides by $\varepsilon(\mathbf{r})$ and obtain:

$$\nabla \cdot \left(\varepsilon(\mathbf{r})\nabla\varphi(\mathbf{r})\right) + \sum_i \frac{z_i^2 e^2 C_{\infty,i}}{k_BT}\varphi(\mathbf{r}) \approx \rho(\mathbf{r}) \tag{7}$$

Using the PB equation, the potentials at distinct grid points surrounding the residue are then calculated for two different protonation states of that residue, protonated and deprotonated. The solver then infers the Boltzmann distribution based on these potentials, describing the probability of finding each of the two considered protonation states. This probability of a state is then translated into a ΔpKa, as done previously[20,52]. Briefly, the difference between the states in the protein is compared with the difference between the states in water, which produces a ΔΔG. This ΔΔG allows us to calculate the ΔpKa. The value of the pKa in the protein is obtained by adding this ΔpKa to the known experimental value in water.

## Limitations of the pKa calculations

Some limitations of the pKa-based approach are listed here. The first part of this section concerns the approximation of the electrostatic work in PBEQ. The linearization mentioned in Eq. 5 supposes that $x \ll 1$, where $x = \left|\frac{z_i e\varphi(\mathbf{r})}{k_BT}\right|$. Since the next term in the Taylor expansion would be $x^2$ and all

further terms are significantly smaller, the linearization always underestimates the magnitude of the potential. At the temperature of 310 K used in this work, $\frac{k_BT}{e} \approx 27$ mV, rendering the approximation valid when $|z_i\varphi(\mathbf{r})| \ll 27$ mV or 0.62 kcal/mol. Because it harbors tight salt bridges and divalent ions, our transmembrane protein system harbors cases with interaction energy above this threshold. For illustrative purposes, we apply this here to a fictive case involving a salt bridge formed by monovalent residue sidechains as is commonly observed in proteins.

If we focus on a pairwise interaction between two (point) charges, their energy is described by the Coulomb law:

$$\Delta E = \frac{z_i z_j e^2}{4\pi\varepsilon_0\varepsilon_r r} \tag{8}$$

Assuming hydrated carboxylate and amino functional groups located at 5 Å from each other, the value of their interaction energy is $\approx -0.852$ kcal/mol. It follows that the potential felt by one of the two partners, e.g., the acidic group $-1$ with zi = -1 is $\varphi = \frac{\Delta E}{z_i e}$. Incorporation of this term into (4) yields the exponential term $e^{\frac{-\Delta E}{k_BT}}$. Computing this exponential would provide the correct value of this fictious case. We can compare it with the approximation returned by the PBEQ solver. We will additionally show how the error would rapidly vanish when applying second and third order approximations of the Taylor expansion.

$$e^{-x} = 1 - x + \frac{x^2}{2!} - \frac{x^3}{3!} + \cdots \tag{9}$$

Using the interaction energy mentioned above yields dimensionless values of 4.207, 2.437, 3.469, and 3.964 when evaluated using the exponential function and its first-, second-, and third-order Taylor approximations, respectively, which shows that the magnitude of the error introduced by the simplification decreases with each added term, from 42% to 18%, and finally to 6%. We now estimate the error, for example of the first-order Taylor approximation, in terms of pKa by calculating $\frac{1}{2.303} \times Ln(\frac{4.207}{2.430}) \approx 0.24$ pKa units, which could be compared with the errors shown in Fig. 1. To save computational resources, our work involved the tailorized, linear approximation. Since the PBEQ solver does not provide the possibility to use a higher-term Taylor approximation but can solve the equation analytically at higher computational costs, we would recommend the analytical approach to users having access to large resources.

The second part concerns the ensemble-based treatment of solvation effects. Hydration and dehydration events were shown previously for their importance in, e. g., protein folding[53,54]. Since the solvent is treated implicitly, Poisson–Boltzmann solvers cannot, in principle, describe such events accurately when they involve buried ionizable residues or polar groups. Some studies attempt to address this limitation by allowing the solvent dielectric constant to vary with ionic strength and concentration[55], or by replacing the uniform protein dielectric with a spatially distributed one estimated from structural features[56]. Other approaches combine both strategies[57]. More advanced alchemical methods include explicit-solvent free energy perturbation and thermodynamic integration, in which a residue is gradually protonated or deprotonated in a fully atomistic environment. These methods require multiple MD simulations at intermediate values of a coupling parameter λ that interpolates between protonated and deprotonated states, typically producing ten or more simulations per residue. Given the workload, such techniques are best suited for systems with fewer candidate pH-sensing residues than in the present case. Some constant-pH simulations treat the solvent explicitly. These generally alternate between conformational sampling and a decision step that determines whether to protonate or deprotonate a residue. This step currently relies on implicit solvent models, such as Generalized Born (GB) or Poisson–Boltzmann approximations, similar to our strategy.

Here, we use an approach that relies on ensemble-averaged structures to approximate solvent effects. Short MD simulations in explicit solvent are

performed, and representative frames are extracted. In these, water molecules can form or break hydrogen bonds, hydrate or dehydrate charged or polar groups, and modulate side-chain conformations and ion positions. These interactions leave a structural and electrostatic imprint on the protein, observable in hydrogen bond patterns, side-chain rearrangements, and nearby ion shifts. Although water molecules are removed before PBEQ calculations, their transient effects are encoded in the frame geometry and charge distribution. Averaging PBEQ outputs over these solvent-conditioned conformations allows us to indirectly account for hydration and dehydration events that would otherwise be missed using a single static structure.

The third part concerns the membrane structure in the computations. The standard implementation of the Poisson–Boltzmann solver typically assigns a uniform dielectric constant to the lipid bilayer, despite its structurally heterogeneous nature. In the present work, we partially addressed this limitation by incorporating spatially resolved membrane features into the dielectric model. Specifically, for each structural snapshot, we quantified membrane thickness using the positions of phosphorus atoms from both leaflets and determined the bilayer midplane from the Z-coordinates of the terminal methyl groups. This enabled us to assign distinct dielectric values to the hydrophobic core and the hydrophilic headgroup regions. Our system includes a membrane composed of a single lipid species, in contrast to the compositional complexity of biological membranes. As the solver only considers the thickness of the different dielectric regions (headgroups and core), discrepancies between the modelled and experimental membrane compositions could lead to PBEQ-specific pKa inaccuracies, particularly if the structural thickness of these regions differs significantly[28]. A more biologically representative membrane composition would likely be beneficial in long, unbiased simulations. However, given the deliberately short duration of the present simulations, it remains uncertain whether incorporating a complex membrane would yield significant differences.

## Strategy of pKa calculations

We used here an ensemble-informed, iterative pKa-calculation strategy, as outlined in the introduction. The pKa of a given residue depends on the electrostatic interactions with neighboring residues. For each structural model (closed, open and desensitized), the pKa calculation was performed in several rounds. In the first round, protonation was set according to the default pKa (Asp, 3.90; Glu 4.07; His, 6.04; Lys, 10.54). A 10 ns long simulation was conducted under this condition, and frames extracted at 0, 2, 4, 6 and 8 ns for pKa calculations using the state-of-the-art CHARMM Poisson-Boltzmann (PB) equation solver, as described below in detail. Some acidic residues are likely protonated, and some basic residues deprotonated at the relevant pH (here < pH7.4 for closed, and < pH5.5 for open and desensitized state). In each subsequent round, the Asp, Glu and His residues with the highest calculated pKa (above the relevant pH) were protonated, while Lys residues with the lowest pKa, below the relevant pH, (as defined above) were deprotonated; the pKa values of the remaining residues was recalculated at the 5 time points of the MD simulation, as described above. In each subsequent round, the MD simulation was started from the coordinates of the initial model. The pKa values plotted in Fig. 1 are from the highest round, or for residues whose protonation state had been adjusted, the last round before this adjustment. Supplementary Fig. 8 shows for selected residues, how the pKa values evolved along the rounds of protonation. The values presented in Fig. 1 correspond to the average of the five time points, measured on three subunits ($n = 5 \times 3 = 15$). These calculations were performed at 'Curnagl', the 96 node HPC cluster based on AMD Zen2/3 CPUs offered by the University of Lausanne, Switzerland, using the container singularity version 3.7.4 to run CHARMM. Further data handlings were performed using in-house Python scripts. The length of the simulation was chosen to be sufficiently long to induce some relaxation of the biomolecules while staying short enough to avoid large conformational changes of the protein. For each frame, the following features were documented, in addition to the protein structure, namely its level of insertion in the membrane, the width of the membrane, and the location of positive and negative ions located within 6 Å of the protein. The convention is to inform any PB solver, which uses implicit solvent and membrane, about the protein insertion. We questioned whether the ionic concentration, set in this work at 0.15 M, suffices to describe the solvent properties in case of frames originating from an MD simulation. Indeed, a frame extracted from a simulation may harbor ions located near the protein, which may cause strong electrostatic interactions not taken into account in an implicit solvent approach. Using a few randomly selected frames, we investigated residues located at the surface of the protein and having neighboring ions located at various distances between 2.5 and 11 Å. Their pKas were then estimated instructing either the protein alone or the protein together with the ion structure. These two calculations allowed to derive a ΔpKa, the difference between the two results, which is expected to be zero if the omission of a given ion does not affect the results. Ions located within the above-mentioned threshold should not be ignored, as lacking this information resulted in a significant pKa shift between 1 and 2.5 units of pKa (Supplementary Fig. 9). Although the Coulomb equation ignores the effect of the solvent considered by the Debye factor, a fit following the Coulomb law sufficed to fit these ΔpKa ($n = 12$, $p < 0.001$, R = 0.9), since our goal was only to determine the distance at which ions may be ignored. We then decided to incorporate the ion screening effect by mentioning explicitly the coordinates of monovalent ions located at less than 3 Å of the protein. This distance was increased to 6 Å for the divalent ions.

Other adjustable PBEQ parameters were set as follows. The dielectric constants of the reference environment, membrane core, membrane headgroups, protein interior and solvent were set to 1, 4, 8, 8, and 80, respectively. Two successive electrostatic estimations with grid spacing of 1.5 and then 0.5 Å were conducted. The PBEQ module from the CHARMM suite treats the non-protein areas of the system implicitly but allows for the assignment of specific permittivity values to the membrane core and headgroups. It does not, however, consider the heterogenous nature of biological membranes. Whereas the thickness of the headgroup region was set to 2 Å in all cases, the exact thickness of the membrane and the center of the membrane along the axis normal to the membrane were calculated for each individual frame. Confirming that this detailed and labor-demanding work has a significant effect on the estimated pKas, we show that the open structure embedding membrane is significantly thicker by about 1 Å than in the case of the closed structure. Ignoring these membrane dynamics would have impacted the pKa calculation of residues located near the membrane headgroups. Conversely, we showed that the membrane thickness did not evolve over time during these simulations.

## Details of the MD simulations

The atomic models of the channel were initially constructed based on the 5WKU[45], 4NTW[58], and 4NYK[59] crystal structures, which are assumed to correspond to a closed, open and desensitized state of chicken ASIC1 channels. These structures appeared to be the most reliable structures when we started the study. Using SWISS-MODEL, homology constructs of hASIC1a, including the generation of short missing loops, were constructed[60]. Chicken ASIC1 shares 90% sequence homology with hASIC1a. Since the large N- and C-termini were not resolved in these structures, they are also absent from our work. Chicken ASIC1 structures harboring the re-entrant N-terminus were recently published in the closed and desensitized states[61]. Due to the lack of an open structure harboring these residues, our study does not include this re-entrant loop. Since this loop does not affect the ASIC1a ectodomain structure, the absence of this structural element does most likely not affect the pKa calculations of extracellular residues. Using the CHARMM-GUI web service[62], each model was inserted in a 1-palmitoyl-2-oleoyl-sn-glycerol-3-phosphocholine (POPC) bilayer (≈ 230 molecules) and solvated (≈ 55000 water molecules) at 0.15 M NaCl. All-atom MD simulations were performed with the CHARMM36 force field[63] using the GROMACS package, version 2021.5[64,65]. The TIP3P water model was used[26]. Bond and angle lengths involving hydrogen atoms were constrained using the LINCS algorithm[27], allowing an integration time step of 2 fs. Short-range electrostatics were cut

off at 1.2 nm. Van der Waals interactions were calculated explicitly up to 10 Å, beyond which a switch function was used to smoothly decrease the interaction force values to reach zero at 12 Å. For long-range electrostatic interactions, the PME algorithm was used[66]. The protein, lipids, and solvent were coupled each to a separate temperature bath with the Nose-Hoover method with a time constant of 1.0 ps[67,68]. The isothermal-isobaric conditions were further ensured by maintaining the pressure around a value of 1 bar via semi-isotropic Parrinello-Rahman coupling.

### Biophysical properties of residues at putative pH sensor locations

When more than four different variants of a specific residue were investigated for their effect on the $pH_{50}$ and $pHD_{50}$, we estimated which side-chain property may likely be most important for the role of the residue. To do so, we performed linear regressions between a specific side chain biophysical property and the measured $pH_{50}$ or $pHD_{50}$. The following lines list the investigated biophysical properties and their source (publication, website). Studied were: the flexibility index[69], the size[70], the hydropathic index according to Kyte-Doolittle[71], the polarity[72], the surface area[73], the bulkiness[74], the propensies of being found in α-helices, β-turn, β-sheet or in coils[75] and the number of hydrogen bonds as donor or acceptor (https://www.imgt.org/IMGTeducation/Aide-memoire/_UK/aminoacids/charge/). The results are summarized in Table 2 and the individual regression analyses are reported in the Supplementary Data 1 file.

### Site-directed mutagenesis and in vitro transcription

The human ASIC1a clone (GenBank U78181[5], containing the mutated Asp212 residue corrected to Gly[76]) and derived mutants were subcloned in the pSP65-derived vector pSD5 that contains 5' and 3' non-translated sequences of β-globin to improve the RNA stability in *Xenopus laevis* oocytes. Most of the individual and combined mutants were generated by Genscript. Several individual mutants were generated in our laboratory by site-directed mutagenesis using the QuikChange approach, with KAPA HiFi HotStart PCR polymerase (KAPA Biosystems). NucleoSpin Plasmid (MACHEREY-NAGEL) was then used to isolate high-copy plasmid DNA from *E. coli*, the mutations were verified by sequencing (Microsynth) and the mMESSAGE mMACHINE kit (Thermo Fisher Scientific) was used for RNA transcription.

### *Xenopus laevis* oocyte preparation and use

To collect oocytes, a small incision on the abdominal wall was performed on female *Xenopus laevis* frogs anesthetized using 1.3 g·L$^{-1}$ of MS-222 (Sigma-Aldrich). The lobe containing the oocytes was surgically removed and then treated with 1 mg·ml$^{-1}$ collagenase for isolation and defolliculation. Healthy stage V and VI oocytes were selected, injected with 50 nl of cRNA at 5–1100 ng·μL$^{-1}$, and stored in Modified Barth's saline containing (in mM) 85 NaCl, 1 KCl, 2.4 NaHCO₃, 0.33 Ca (NO₃)₂, 0.82 MgSO₄, 0.41 CaCl₂, 10 HEPES and 4.08 NaOH at 19°C. All animal experiments were carried out in accordance with Swiss laws and approved by the animal welfare service of the Canton de Vaud.

### Electrophysiological measurements

Electrophysiological recordings were performed 24–72 h after cRNA injection. Whole-cell currents were recorded by two-electrode voltage-clamp using two glass electrodes with a resistance < 0.5 MΩ when filled with 1 M KCl. A Dagan TEV200 amplifier (Minneapolis, MN) and an Instru-TECH LIH 8 + 8 interface were used. The potential was held at −60 mV during the entire protocol. The current was measured with a sampling interval of 20 ms and filtered at 2 kHz. PatchMaster software (HEKA-Harvard Bioscience) was used to perform recordings. A fast perfusion system (cFLow 8 channel electro valve unit (CellMicroControls)) was used to perfuse the oocytes with solutions by gravity at a flow rate of 8–12 mL·min$^{-1}$. Once per minute, the oocytes were exposed to a stimulation solution with a selected pH for 5–10 s. The pH-current curves were fitted to the Hill equation: $I = I_{max}/[1 + (10^{-pH_{50}}/10^{-pH})^{nH}]$, where $I_{max}$ is the maximal

current amplitude, $pH_{50}$ is the midpoint of the activation curve and $n_H$ is the Hill coefficient; SSD curves were fitted to an analogous equation. The recording solution contained (in mM) 110 NaCl, 2 CaCl₂, 10 HEPES for pH ≥ 6.8 (10 MES instead of HEPES for pH < 6.8). NaOH or HCl were used to adjust the pH.

### Statistics and reproducibility

For the analysis, currents were normalized to the maximum peak current. For fits and statistics, Graphpad Prism (version 10) was used. For each data group, a D'Agostino and Pearson normality test was performed. A one-way ANOVA test followed by a Dunnett's multiple comparisons test was used when < 25% of groups showed a non-normal distribution, and a Kruskall-Wallis test followed by a Dunn's multiple comparison test was used when ≥ 25% of groups showed a non-normal distribution. Statistical tests were two-sided. Each experiment was carried out on at least two different experimental days and in batches of oocytes from different frogs. Individual data points are shown in all figures and mean ± SEM is also plotted.

The UCSF chimera software was used for structural images and to measure the distance between different atoms.

### Reporting summary

Further information on research design is available in the Nature Portfolio Reporting Summary linked to this article.

## Data availability

All computational and experimental data are contained in the article and in the supplementary material. The Supplementary Data 1 file contains details of the correlation analysis. The Supplementary Data 2 file contains the source data. All other data are available from the corresponding author on reasonable request.

## Code availability

Custom scripts were used for input preparation and data analysis, but no novel code was developed.

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

## Acknowledgements

The MD simulations were made possible by computational resources allocated by the Swiss National Supercomputing Center (CSCS) under the project IDs sm12, sm20, s968, s1037, s1223 and s1234. The pKa calculations were performed using Curnagl cluster, a 96 node HPC cluster based on AMD Zen2/3 CPUs maintained by DCSR team at the University of Lausanne, Switzerland. The Charmm Software was used for the pKa calculation[77]. OB expresses gratitude to Simon Bernèche for valuable discussions on the Poisson-Boltzmann equation. This work was supported by grant 310030_207878 from the Swiss National Science Foundation to S.K. We thank Eleanor J McKay for correcting the manuscript and Eleonor Centonze, Marc Bohnet Mina Hanna and Miguel van Bemmelen for comments on the manuscript.

## Author contributions

All authors reviewed and edited the manuscript. S.K., O.B. and O.M. conceived the project and designed the computational and functional approaches. I.G. and O.M. carried out the functional experiments. O.B. did the computational work. All authors contributed to the data analysis and figure preparation. O.M., O.B. and S.K. wrote the manuscript.

## Competing interests

The authors declare no competing interests.
