## [Transparent Peer Review file · Communications Biology]

Molecular mechanisms and hotspots of pH sensing in ASIC1a revealed by computational and functional analysis

Corresponding Author: Professor Stephan Kellenberger

Version 0:

Reviewer comments:

Reviewer #1

(Remarks to the Author)

The article addresses the extremely important problem of identifying pH sensors in an ASIC channel. Despite numerous efforts, this issue has still not been resolved. In this work, the Poisson-Boltzmann equation is used to predict the pKa values for titratable ASIC1a residues. The predicted pH sensors have been thoroughly tested using the mutagenesis approach. The research is conducted on a highly systematic manner, and the data obtained is very valuable. Without a doubt, the results obtained deserve publication.

My comments relate to the presentation of a large amount of data and the text of the manuscript, rather than the idea of the work and the results.

Introduction. This section is short and describes only the basics of ASICs. The problem of pH sensor identification is described in one sentence "It has previously been shown that mutations of Asp, Glu and its residues located in different extracellular domains of ASIC1a change the dependence on pH. These residues can represent H⁺ sensors or influence the steps following protonation." I recommend expanding this section and describing previous studies (predictions and mutations) in more detail. This will allow readers to understand the motivation of this research.

Results, legend for Fig. 1. "(C-F) Calculated pKas values (meaning SD, n = 8) for titrated ASIC1a residues in humans." Please explain how the n=8 value was obtained for the calculated pKas values.

Results, pKas prediction. Please explain how three specific three-dimensional structures were chosen among numerous available structures.

Results. The main interest of the authors is the pH₅₀ values for activation and steady-state desensitization. However, it is known that some ASIC1a mutations significantly affect the form of the response, in particular, a sustained current component may occur. I suggest mentioning such cases or noting that no drastic changes were found in the response form.

Discussion. A comparison with previously published results is provided in the text, but a more systematic comparison is desirable. Key new results should be highlighted. A list of proposed pH sensors that have not been previously predicted. A list of residues that have never mutated before. A list of hotspots that have not been previously identified.

Reviewer #2

(Remarks to the Author)

In this work, Bignucolo et al. claim to have identified amino acid residues in human ASIC1a responsible for proton-mediated transitions to the open and/or desensitized states. Computational pKa calculations using the Poisson-Boltzmann equation identified 38 potential pH-sensing residues, of which 27 were experimentally validated through mutation-induced shifts in pH dependence. Further mutational analysis confirmed six residues as true pH sensors and revealed functional interactions between several others. Key pH-sensing hotspots were localized in the palm, acidic pocket, thumb, and wrist structural

domains, with specific Lys, His, and Glu residues playing critical roles. The study also investigates how protonation states and electrostatic interactions influence ASIC1a activation and steady-state desensitization.

Overall impression of the work:

The manuscript presents extensive work with many interesting and important findings related to the molecular basis of pH gating in ASIC1a and ion channel regulation by extracellular pH. The conclusions are generally well-supported, and most statements are scientifically accurate. However, the text is overly complex and difficult to read, which affects its clarity and accessibility. A more concise and structured presentation would improve readability. Additionally, there are scientific inaccuracies, missing clarifications, and potential misinterpretations that need to be addressed. Specific issues are outlined in the detailed comments below.

Comments

General

1. Abstract: the authors broadly state that they have identified critical hotspots involved in ASIC1a gating, but they should be more specific about their key findings. In particular, they need to explicitly mention the confirmed residues that may drive the activated and desensitized states.
2. pKa predictions: the manuscript overlooks necessary corrections for accurate pKa predictions. It does not discuss the use of explicit solvent models or the impact of counterion effects on pKa shifts, which are crucial for accuracy.
3. Experimental data presentation: a table summarizing the experimentally determined pH50 and pHD50 values for both control and mutant channels is necessary. Currently, the manuscript does not provide numerical values for these measurements anywhere in the text, which limits data transparency and reproducibility.
4. Results section – pH sensor properties: the section "Biophysical properties of residues at pH sensor locations" does not present actual results but instead describes methodological aspects. This part needs to be rewritten to clearly report the findings rather than focusing solely on the methods.
5. Methods section – Poisson-Boltzmann (PB) equation: the subsection "Derivation of the Poisson-Boltzmann Equation for pKa Calculations" reads like a textbook excerpt. To improve readability, the main formula and its explanation should remain in the main text, while the full derivation should be moved to the Supplementary Materials.
6. The authors do not mention that the PB equation has limitations, such as: a) it assumes a continuum dielectric model, which may not fully capture local solvation effects; b) it does not account for explicit water molecules or molecular flexibility; c) it may require empirical corrections for highly charged environments.
7. The approach of conducting successive MD simulations with progressive protonation states raises questions about convergence and reproducibility. Does the method always converge to the same result, or does it depend on initial conditions? How many iterations does it typically take to achieve stability? Are there cases where this method fails or produces artifacts?
8. Since the study is very extensive, the authors should add a Conclusions section with a concise and clear summary of the key findings, as well as the study's limitations.

Specific

1. Lines 41-42. Please explicitly list the eight different ASIC subunits in the text and provide the corresponding references.
2. Line 56: Add a colon after "states".
3. Line 63: please, include the corresponding references.
4. Lines 78-79: The PB equation solves for the electrostatic potential as a function of charge distribution in an ionic medium, but pKa estimation from PB calculations typically involves additional considerations, such as desolvation effects and reference states.
5. Lines 81-82: The Boltzmann distribution applies to mobile ions in solution rather than the biomolecule itself.
6. Lines 89-91: The electrostatic potential itself does not directly yield free energy changes. Instead, the calculated potential is used to determine the electrostatic contribution to the Gibbs free energy, which is then related to pKa shifts.
7. Lines 117-118: What is the minimum ΔpK_a considered biologically relevant?
8. Line 120: E238 is identified as the highest-ranked pH sensor. Why does it have an extremely low pKa (0.58) in the closed state? This suggests a highly stabilized negative charge, which is unusual for a buried glutamate.
9. Has ASIC1a pKa data been measured experimentally? If no experimental data exist, how reliable is the computational model?
10. Lines 212-265: Residues H73, H78, H173, E219, and D409 were previously classified as part of the palm domain but are now referred to as part of the wrist domain. Please clarify this apparent discrepancy.
11. Lines 302-303: "K374 shares a proton with both E375 and E413". Does this mean delocalized protonation, a dynamic equilibrium, or direct H-bonding?
12. Lines 312-316: The E277 mutations suggest that its titratability affects activation, but pKa values indicate it remains negatively charged in all states. How does a permanently charged residue influence pH gating?
13. Lines 473-474: H73 may drive or stabilize activation? Please, clarify.
14. The Discussion section is very extensive and jumps between computational and experimental findings without always making clear transitions. Please, reorganize findings into clearer parts (e.g., computational predictions, mutagenesis findings, interactions and structural insights).
15. The Discussion section refers to "strong" or "small" effects without quantifying these findings. Please, quantify findings where possible to support qualitative descriptions.
16. Lines 605-606: Protonation states are influenced by pKa, pH, and the local environment (e.g., nearby charged residues, membrane polarity, hydration, etc.), not just pKa and pH alone. Please, clarify the local microenvironmental effects.
17. Lines 624-630: Membrane interfaces do not behave as uniform dielectrics - they have graded dielectric environments (e.g., lipid headgroups vs. hydrophobic core). The presence of charged lipids and embedded proteins can further alter the

electrostatic landscape.

18. Lines 649-650: It should explicitly state that the Boltzmann distribution accounts for mobile ion distributions in an electrolyte solution (not within the protein itself).
19. Lines 653-655: The linearized Poisson-Boltzmann equation assumes weak electrostatic potentials, but this approximation breaks down for highly charged surfaces (such as the interior of an ion channel). You should mention that the linearization is valid only when the electrostatic potential is low (e.g., ≤ 25 mV at room temperature).
20. Lines 661-664: Implicit solvation models do not always accurately capture local hydration effects, which can cause errors in pKa calculations. The Born solvation model or explicit solvent corrections are sometimes needed for accuracy.
21. Lines 690-691: How does the implicit treatment compare to fully explicit solvent approaches? Is there any benchmarking against experimental data to validate the pKa shifts obtained?
22. The study considers ions within 3 Å (monovalent) and 6 Å (divalent) of the protein explicitly while treating the rest implicitly. The chosen distance thresholds seem somewhat arbitrary. Are they based on systematic analysis, or were they optimized to fit experimental pKa shifts?
23. The claim that ignoring ions within this range causes a pKa shift of 1–2.5 units is significant, but is this effect uniform across different protein environments? More validation may be needed.
24. Line 737: Realistic neuronal membranes are not composed solely of POPC – they contain cholesterol, PIP2, and other lipid species that can affect ASIC function. As it is known that ASICs interact with specific lipid domains, was any test done to check if this artificial bilayer affects the protein's stability?
25. Line 738: The 0.15 M NaCl concentration is physiologically reasonable but does not account for other ion types (e.g., Ca^{2+} , K^+) that may affect ASIC1a function. Moreover, the authors themselves emphasize in the Discussion the important influence of Ca^{2+} on pH sensors.
26. Line 740: TIP3P is a common but relatively simple water model with known limitations in describing hydrogen bonding and ion hydration. Why not use TIP4P or TIP5P models, which better represent bulk water properties?
27. Lines 747-748: Semi-isotropic control assumes membrane fluctuations along the lateral plane, but ASICs undergo large conformational changes. Would anisotropic pressure coupling be more appropriate for a membrane protein undergoing structural transitions?
28. Line 753: Linear regression assumes that a single side-chain property has a direct and independent effect on pH50 and pH50. But residue contributions to pH sensing are often nonlinear and context-dependent, and synergistic effects between size, charge, and flexibility are likely, but linear models do not account for these interactions. Were multiple regression models or machine learning approaches tested? If only single-variable regression was performed, potential multicollinearity effects were ignored.
29. Did the authors consider other electrostatic parameters, such as pKa shifts of nearby residues, solvent accessibility, or charge distribution?
30. Were MD simulations used to check if residue mutations introduce local unfolding or changes in side-chain mobility?
31. Did the authors compare results with ASIC1a from other organisms? If similar correlations exist across species, this would strengthen their conclusions.
32. Line 766: Please clarify how 5' and 3' non-translated sequences of β -globin improve the protein stability?
33. Figures 2–6: Clearly indicate the meaning of the dotted lines used in these figures.
34. Figures 2–6: Specify the number of individual cells used for measurements. While Figure 2 mentions $n = 7-15$, some bars (e.g., WT, K374Q, etc.) appear to have more than 15 data points. Please verify and clarify.
35. Figure 5I: In the right panel, the dotted line for pH50 corresponds to 6.7 instead of 6.5, which I assume is the WT control pH50 value. This differs from the placement of the dotted line in all other figures. Please explain this shift of 0.2 units. If this is an error, it may be necessary to reconsider the impact of residue K246 when compared to a pH50 of 6.5.
36. Figure 5H: The dotted line for pH50 is placed at 7.1 instead of 7.15, which is used consistently in all other figures. Please clarify this discrepancy.

Reviewer #3

(Remarks to the Author)

Bignucillio et al., 2025 review

This comprehensive work by Bignucolo et al., conducts a systematic assessment of titratable residues in ASIC1a, driven by pKa estimates, to reinforce and extend existing knowledge about ASIC pH sensing hotspots. The approach, experiments and interpretation are well conducted and justified. This work will be a useful resource for those in the field. Below are my suggestions to enhance the utility of this manuscript.

Major

This project appears to be an updated version for Liechti et al., 2010 JBC. The Liechti paper also used a PB approach to classify residues based on pKa (although limited to fewer structures at the time), followed by functional characterization to identify pH sensors and hotspots. Many (if not most) of the positions identified in the present work were first described by Liechti et al. 2010 or Paukert et al., 2008. It is very important the authors clearly state what new positions or insights arise from this study versus confirmation of past work. In lines 536-541, they discuss other papers that undertook systematic assessments of ASIC1a side chains, but didn't mention the Liechti paper in this context. The authors should directly and explicitly compare this work with the prior Liechti paper to highlight for the reader what new positions, or interactions have been identified and how their present approach has improved (or not) upon that of ~ 15 years ago.

The comprehensive and methodical nature of this data is both a strength and weakness. There is so much here it is difficult to navigate through. Therefore, the authors should pay particular attention to their summary figures. For example, Figure 2I contains a great deal of information but it is too cluttered. The authors use color for both domains, such as blue for thumb and yellow for palm, as well as for the effect size of mutation. The mutations are shown as spheres and the spheres are colored but it is still too confusing. I suggest the authors use color only for the sphere mutations, leaving the cartoon backbone in grey. This should allow the mutations to stand out much better. Also, Figure 7 contains attempts to summarize too many

things to actually be useful. Consider removing the boxes around the descriptions of “putative pH sensors” while leaving those around “confirmed”. This should reduce clutter and increase readability.

In the legend of Figure 7, the authors refer to a position as either “driving” or “stabilizing” the transition based on “changes in protonation fraction”. Table 2 further refers to protonation fractions as being in closed, C->O, O, C-> D or End SSD. How do the authors distinguish pKa’s during a transition as opposed to endpoints?

The methods mention this implementation of the PB approximation considers the extent to which neighboring residues are also protonated (lines 673-675). This is such an important point that the authors should also mention in the Results or Introduction somewhere. The reader shouldn’t have to go digging to answer this fundamental question!

The authors should consider a scatter plot of delta pKa between states versus delta pH50 or pH50 to determine how well the PB prediction predicts sites. They should also consider a scatter plot of delta pH50 versus delta pH 50 to better indicate which side chains influence activation or SSD or both.

Minor

Introduction, Line 54 should include a “the” before “wrist”

Introduction, the PB approach been used in prior work such as Liechti et al. to study protonation. This paper, or others, should be cited for precedent.

Line 244, double mutant cycle analysis is prone to false positives and negatives (see PMID 25311637)

Line 434, “analysis confirmed six residues as true pH sensors”. What is a “true” pH sensor? And how could other residues “contribute critically to pH dependence” (next line) but not be “true” sensors? Some clarification, and also identification of these “true” sensors here is needed.

Figure 3I, H173/E219 and D409 are not the “wrist” domain as the figure title suggests.

Line 438 “provides a robust framework for identifying pH-sensing mechanisms in other ion channels”. Some other channels should be mentioned.

Version 1:

Reviewer comments:

Reviewer #1

(Remarks to the Author)

All my comments are addressed carefully and correctly.

Reviewer #2

(Remarks to the Author)

The authors have addressed the majority of my previous comments and questions. However, a few critical points remain that still require attention:

1. Please clarify the changes made to Fig. 1D. It appears to differ from the previous version. Please verify its accuracy and include confidence intervals in the figure.
2. Line 32: I suggest specifying that the desensitization referred to is “steady-state desensitization” for clarity.
3. Line 44: “ASIC3a, -3b and -3c” applies specifically to human ASIC3, not to ASIC3 isoforms across all mammals. Please revise this accordingly to avoid overgeneralization.
4. Line 676: The phrase “determinators of pH dependence” should be replaced with more standard scientific language, such as “key determinants of pH sensitivity”.
5. Conclusion: I recommend closing the paragraph with a forward-looking statement that highlights how these insights could guide future investigations into other ASIC isoforms, or that places the findings in a broader physiological or pharmacological context, for example, by mentioning potential implications for ASIC-related diseases or therapeutic development.

Reviewer #3

(Remarks to the Author)

The authors have done substantial revisions to improve clarity and presentation. While this is still a huge amount of work, and difficult to parse through, I do not have any further suggestions to improve. Here are some minor thoughts:

In reading the response to R1, the authors state they have an n = 15 or 8. But they describe this as at each amino acid they measure protonation at 3 subunits and 5 time points, thus giving n = 15. I disagree with this interpretation. N is commonly treated as independent replicates. Counting the same amino acid at multiple points in time is not independent. The assumption that each subunit is independent is also shaky but less concerning.

Intro, line 80 uses “based” twice in the same sentence.

Figure 1, in panel D, the symbols within a column are in the following order: closed, open and desensitized. In the other panels, the order is open, desensitized, closed. Why the difference?

Figure 5, panels B – D, please double check the use of K211D versus K211E in these panels. The ms text, figure legend and figure labels might be conflicting at times.

Version 2:

Reviewer comments:

Reviewer #2

(Remarks to the Author)

The authors have addressed all my comments and questions.

Reviewer #3

(Remarks to the Author)

The authors have satisfactorily addresses all my concerns.

Revision Bignucolo et al: Response to reviewers

Dear reviewers,

First, we would like to thank you for the insightful comments. We revised the manuscript according to the comments and hope that it is now acceptable.

The most important changes of the manuscript are:

- Previous work on this topic is discussed in the introduction. Previous work is summarized in Table 1 and supplementary Table 1.
- We have extended the description of the computational approach (Results and Methods); some of it has also been re-placed into the introduction; Supplementary Table 4 and Figures 8 and 9 contribute further to this description.
- The discussion has been completely re-organized

I would like to mention that during the revision, we identified a few errors in the presented pKa and pH50 values, that have been corrected. These changes did in no way affect the conclusions of the manuscript.

Sincerely,
Stephan Kellenberger

Reviewers' comments:

Reviewer #1 (Remarks to the Author): Ion channel regulation and function, Physiology and Biochemistry

The article addresses the extremely important problem of identifying pH sensors in an ASIC channel. Despite numerous efforts, this issue has still not been resolved. In this work, the Poisson-Boltzmann equation is used to predict the pKa values for titratable ASIC1a residues. The predicted pH sensors have been thoroughly tested using the mutagenesis approach. The research is conducted on a highly systematic manner, and the data obtained is very valuable. Without a doubt, the results obtained deserve publication.

My comments relate to the presentation of a large amount of data and the text of the manuscript, rather than the idea of the work and the results.

Introduction. This section is short and describes only the basics of ASICs. The problem of pH sensor identification is described in one sentence "It has previously been shown that mutations of Asp, Glu and its residues located in different extracellular domains of ASIC1a change the dependence on pH. These residues can represent H⁺ sensors or influence the steps following protonation." I recommend expanding this section and describing previous studies (predictions and mutations) in more detail. This will allow readers to understand the motivation of this research.

Response: We have extended the discussion of previous studies in the introduction (see lines

64-78). We also provide a detailed table listing the effects of mutations in previous and in the present study (Table 1 and Supplemental Table 1). To avoid repetitions, we discuss the overlap between this and previous studies in the discussion part and refer to the tables. The analysis shows that most of the residues that we have identified here as involved in ASIC pH dependence have already been mutated in previous studies investigating activation. The earlier data come however from a large number of studies carried out under different conditions. The strength of our study is the comprehensive approach and the strategy of testing mutations to several amino acids of residues of interest. Regarding SSD, information on structure-function aspects of SSD has so far come almost exclusively from our laboratory.

Results, legend for Fig. 1. "(C-F) Calculated pKas values (meaning SD, n = 8) for titrated ASIC1a residues in humans." Please explain how the n=8 value was obtained for the calculated pKas values.

Response: In the revised manuscript we improved the description of the analysis (see lines 839-857). Briefly, for a given structure (i.e. closed, open or desensitized), several rounds of protonation were carried out. In each round, MD simulations of 10ns were carried out and pKa values were calculated at 0, 2, 4, 6 and 8 ns. Therefore, we obtained for each residue $5 \times 3 = 15$ values (5 time points x 3 subunits), and for residues of which protonation/deprotonation can occur at two atoms (i.e. the two oxygen atoms of a Glu side chain), the n is even=30. The "n" has now been corrected to 15. The legend has been adapted.

Results, pKas prediction. Please explain how three specific three-dimensional structures were chosen among numerous available structures.

Response: These were the most reliable structures when the project was started. In the revised manuscript, this is stated under "Details of the MD simulations", lines 901-904.

Results. The main interest of the authors is the pH50 values for activation and steady-state desensitization. However, it is known that some ASIC1a mutations significantly affect the form of the response, in particular, a sustained current component may occur. I suggest mentioning such cases or noting that no drastic changes were found in the response form.

Response: In the revised manuscript we provide a figure (supplementary Fig. 2) showing the sustained current / peak current ratios for all neutralization mutants. This ratio was only affected by mutations at a few specific positions. This is now discussed at the end of the paragraph " Measurement of the ASIC1a pH dependence of activation and SSD", lines 185-187.

Discussion. A comparison with previously published results is provided in the text, but a more systematic comparison is desirable. Key new results should be highlighted. A list of proposed pH sensors that have not been previously predicted. A list of residues that have never mutated before. A list of hotspots that have not been previously identified.

Response: In the revised manuscript we show better what is novel in this study. In fact, most of the residues that we identify here as being pH sensors of activation or just important for ASIC activation have been reported in previous individual studies. What is novel in our study for activation is the comprehensive approach designed not to miss any pH sensor, as well as the in-depth analysis of sites of interest by mutations to several different amino acid residues, to determine whether a given residue that affects pH dependence is likely a pH

sensor or not. An additional novelty is the comprehensive analysis of several hotspots of pH sensing, as K374/E375/E314 and E242/K246/D409.

Regarding structure-function of SSD pH dependence, there exist however practically no other studies than ours, and our previous study (Liechti et al.) was much less exhaustive. To provide an overview of the findings and to distinguish our from previous data, we provide a table (Table 1) that contains key residues of activation in our study and lists all previous results on these positions. In addition, in order to analyze the risk of having missed any residue, we provide supplementary table 1, which shows for all extracellular and transmembrane titratable residues that were not functionally investigated in the present study 1) any functional information from previous studies, and, 2) for the titratable residues that have never been functionally investigated, a short structural interpretation. With this, the discussion has been thoroughly re-organized. Most of the text concerning this aspect is found lines 64-78 and 603-607. We also highlight the novelties of our study in the discussion, on lines 613-623.

Reviewer #2 (Remarks to the Author): ASIC Channels, structure and function, modulators

In this work, Bignucolo et al. claim to have identified amino acid residues in human ASIC1a responsible for proton-mediated transitions to the open and/or desensitized states. Computational pKa calculations using the Poisson-Boltzmann equation identified 38 potential pH-sensing residues, of which 27 were experimentally validated through mutation-induced shifts in pH dependence. Further mutational analysis confirmed six residues as true pH sensors and revealed functional interactions between several others. Key pH-sensing hotspots were localized in the palm, acidic pocket, thumb, and wrist structural domains, with specific Lys, His, and Glu residues playing critical roles. The study also investigates how protonation states and electrostatic interactions influence ASIC1a activation and steady-state desensitization.

Overall impression of the work:

The manuscript presents extensive work with many interesting and important findings related to the molecular basis of pH gating in ASIC1a and ion channel regulation by extracellular pH. The conclusions are generally well-supported, and most statements are scientifically accurate. However, the text is overly complex and difficult to read, which affects its clarity and accessibility. A more concise and structured presentation would improve readability. Additionally, there are scientific inaccuracies, missing clarifications, and potential misinterpretations that need to be addressed. Specific issues are outlined in the detailed comments below.

Comments

General

1. Abstract: the authors broadly state that they have identified critical hotspots involved in ASIC1a gating, but they should be more specific about their key findings. In particular, they need to explicitly mention the confirmed residues that may drive the activated and desensitized states.

Response: In the revised manuscript, the residues identified as "confirmed pH sensors" for activation and SSD are listed.

2. pKa predictions: the manuscript overlooks necessary corrections for accurate pKa predictions. It does not discuss the use of explicit solvent models or the impact of counterion effects on pKa shifts, which are crucial for accuracy.

Response: We agree that these are two important aspects of the pKa predictions. In the revised manuscript we have improved the discussion of these aspects. Here we discuss each aspect separately.

Solvent: To our knowledge, there are not yet explicit solvent models that can be used with the resources at our disposal. The methods currently available to us include explicit free energy perturbation combined with thermodynamic integration, QM/MM approaches, or lambda-based constant pH molecular dynamics simulations, all of which are, in our view, too computationally demanding for the resources we have access to. There are also “non-lambda-based” constant-pH MD simulations with explicit solvent (AMBER cpHMD, NAMD cpHMD Roux, and a recent GROMACS implementation). As we understand them, these methods separate the process into two phases: a classical MD production phase, which nowadays typically uses explicit solvent, and a “decision” phase of various forms, during which one or several residues may undergo a protonation change. The calculations required for this decision step are performed using implicit solvent (typically GB or GB/PB approximations), similar to our approach. As a compromise between the scale of our protein, containing many potential pH-sensing residues, and the moderate computational resources available, we designed a strategy that, in our view, partially accounts for solvent effects. Our approach relies on successive short MD simulations, each followed by updated pKa calculations, leading to ensemble-based pKa estimations. Although the solver does not “see” the explicit water molecules, the MD simulations capture the influence of solvation implicitly, such as side-chain displacements driven by hydrogen bond formation or disruption. These rearrangements have a significant impact on local electrostatics, reflect hydration states in a probabilistic manner (albeit over a limited number of frames), and help mitigate errors arising from single-snapshot artifacts. We believe that, despite several limitations, this strategy represents a meaningful improvement over directly submitting a static crystal structure to the solver. This is discussed in the revised manuscript under "Limitation of the pKa calculations", lines 791-818.

Counterions. We asked ourselves the same at the beginning of the project, especially because the short MD simulations are expected to bring ions close to the channel in a stochastic manner, unlike the experimental conditions that led to the crystal structures. We already reported this questioning and strategy in the method section of the original manuscript. Briefly, our calculation considers ion screening effects for monovalent ions within 3Å and divalent ions within 6Å of the protein (Lines 868-883). We have extended this discussion with supplementary Fig. 9 that shows the distance dependence of the screening effect of monovalent ions. To draw attention to this correction outside the method section, we refer to inclusion of counter ions for the pKa calculation at the end of the introduction section (lines 83-101).

3. Experimental data presentation: a table summarizing the experimentally determined pH50 and pHD50 values for both control and mutant channels is necessary. Currently, the manuscript does not provide numerical values for these measurements anywhere in the text, which limits data transparency and reproducibility.

Response: All this numerical information was already in the initial version present in the source data file. We agree that we should provide this information in a more reader-friendly way. We include in the revised manuscript such a table. Since this table is quite big, we include it in the supplemental material as Suppl. Table 2. We refer to it in the legend to Fig. 2 and in the main text.

4. Results section – pH sensor properties: the section "Biophysical properties of residues at pH sensor locations" does not present actual results but instead describes methodological aspects. This part needs to be rewritten to clearly report the findings rather than focusing solely on the methods.

Response: We realize that the title of this section was misleading. This section describes the approach for testing for the correlation of the pH50 shift at residues of which ≥ 4 mutations were tested with a set of side chain properties. We are convinced that it makes sense to present the results of these analyses together with the presentation of the functional analysis of titratable residues of different domains (as opposed to presenting them all together in this section. In the revised manuscript we have adapted the title and parts of the text of this section (see lines 236 ff).

5. Methods section – Poisson-Boltzmann (PB) equation: the subsection "Derivation of the Poisson-Boltzmann Equation for pKa Calculations" reads like a textbook excerpt. To improve readability, the main formula and its explanation should remain in the main text, while the full derivation should be moved to the Supplementary Materials.

Response: We believe that this derivation plays an important role in the manuscript and would be better retained in the main text rather than moved to the supplementary material. To our knowledge, this kind of pedagogical explanation is not available in standard textbooks. The section was developed by Olivier Bignucolo in collaboration with Simon Bernèche (see acknowledgements), with the intention of offering a balanced presentation—simplified enough to be accessible to biologists and general readers, yet rigorous enough to convey the underlying principles of the Poisson–Boltzmann approach. Our goal was to bridge the gap between overly simplified accounts and technically dense treatments, making the material more approachable to a broader audience.

6. The authors do not mention that the PB equation has limitations, such as: a) it assumes a continuum dielectric model, which may not fully capture local solvation effects; b) it does not account for explicit water molecules or molecular flexibility; c) it may require empirical corrections for highly charged environments.

Response: In the revised manuscript we have included a discussion of the limitations of the PB approach. We added a more complete description placing our approach between less accurate/powerful and computer time-economic tools and more accurate but computationally more demanding algorithms like TI, QM/MM. We also discuss the specificities of constant pH MD simulations in this context. This can be found in the second paragraph of the Discussion (lines 501-513) and the Methods section (lines 757-836).

7. The approach of conducting successive MD simulations with progressive protonation states raises questions about convergence and reproducibility. Does the method always converge to the same result, or does it depend on initial conditions? How many iterations does it typically take to achieve stability? Are there cases where this method fails or produces artifacts?

Response: We realize that the description of the MD simulations was not sufficiently clear in the submitted manuscript. Briefly, for a given structure (i.e. closed, open or desensitized), several rounds of protonation were carried out. In each round, MD simulations of 10ns were carried out and pKa values were calculated at 0, 2, 4, 6 and 8 ns. In each round of protonation, the MD simulations were started from the original corresponding structural model (i.e. closed, open or desensitized), with only the protonation of some residues adapted, based on the pKa calculations of the previous round. At the initial conditions, the titratable residues are protonated/deprotonated according to their "bulk" pKa. This protonation/deprotonation state will influence the pKa calculation of residues that are in their proximity. In subsequent rounds, residues are progressively protonated/deprotonated according to their calculated pKa, which will in turn influence the pKa of neighboring residues. Changes in pKa of a given residue occur mostly after a protonation/deprotonation of residues in their proximity. This is now clarified in the methods (lines 809-818 and 838-857). In the simulations, the three subunits behave not always in the same way, and as consequence, the pKa values and protonation status at some positions differed between subunits. The values provided in Fig. 1 are mean values of the 5 times points x 3 subunits as described on lines 839-846. As initial condition we always used the "bulk pKa value" (i.e. 3.86 for Asp, etc.). The observation that for many positions, the calculated pKa values were substantially different from the initial value, suggests that the initial value did not limit the pKa calculation. Currently we have no alternative method with equal reliability for pKa calculation, therefore we cannot test the reliability of the method. If the method provides false positive results, the functional analysis will not confirm the pH-sensing role of the residue, and therefore the given residue will be rightly excluded from the list of pH sensors. The biggest risk is that the pKa calculation does not identify an important pH sensor. In supplementary table 2 of the revised manuscript, we list all titratable residues with indication of functional studies made, and estimate, from structural information, the potential of pH sensing for each residue that has not been functionally studied. With this analysis we are positive that we have not missed important pH sensors. In Supp. Fig. 8 we show for a few selected residues how their pKa value evolved over the rounds of protonation. This is discussed in the Methods on lines 838-857.

8. Since the study is very extensive, the authors should add a Conclusions section with a concise and clear summary of the key findings, as well as the study's limitations.

Response: A conclusion section has been introduced in the revised manuscript.

Specific

1. Lines 41-42. Please explicitly list the eight different ASIC subunits in the text and provide the corresponding references.

Response: The subunits are mentioned, and references are provided in the revised manuscript.

2. Line 56: Add a colon after "states".

Response: change made

3. Line 63: please, include the corresponding references.

Response: Based on your comment and a comment by reviewer #1, this passage has been extended, including more information on the individual studies and the corresponding references (see lines 64-78 in the revised manuscript).

4. Lines 78-79: The PB equation solves for the electrostatic potential as a function of charge

distribution in an ionic medium, but pKa estimation from PB calculations typically involves additional considerations, such as desolvation effects and reference states.

Response: We concur with the reviewer that these introductory lines were overly summarized and may give the incorrect impression that solving the PB equation alone is sufficient to obtain the pKa values. We have extended and clarified the computational approach, as shown on lines 83-101.

5. Lines 81-82: The Boltzmann distribution applies to mobile ions in solution rather than the biomolecule itself.

Response: In our understanding, it applies to both, but differently. As for the mobile ions, given that the solvent is treated implicitly, we think that the solver attributes a charge density as a function of the medium concentration and the static charges present nearby the biomolecule. As for the protein, the partial and full charges are taken from the PSF and topology/parameter files, and their position from the uploaded structure file. This is discussed on lines 121-123.

6. Lines 89-91: The electrostatic potential itself does not directly yield free energy changes. Instead, the calculated potential is used to determine the electrostatic contribution to the Gibbs free energy, which is then related to pKa shifts.

Response: We agree with the reviewer. Since the method section describes the calculation more in detail and to avoid repetition, we removed this sentence and refer the reader to the Method section.

7. Lines 117-118: What is the minimum ΔpK_a considered biologically relevant?

Response: We use here the ΔpK_a of a given residue between structural models (for example between the closed and open structural model) as an indication that the environment of this residue changed between states. Such a change in environment would be expected if the residue changes its protonation status during the transition. For example, if lowering of the pH to a closed ASIC protonates a given residue and this residue contributes to channel opening, this protonation will induce conformational changes close to the residues which will change its environment, and with this likely its pKa. The more important criterium for a given transition is whether, based on the pKa values and the change in pH inducing this transition, this residue changes its protonation state. We have estimated the relevance of each titratable residue for a transition based on the two aspects, the range of the pKa values, and on whether they changed between states, as indicated on lines 148-160. These parameters were considered for an initial ranking without attributing a threshold. The functional analysis would then show whether mutations of residues induced significant changes in pH dependence.

8. Line 120: E238 is identified as the highest-ranked pH sensor. Why does it have an extremely low pKa (0.58) in the closed state? This suggests a highly stabilized negative charge, which is unusual for a buried glutamate.

Response: This is an intriguing point. We would expect a lower pKa in the closed as compared to the open and desensitized conformation, in which the acidic pocket is collapsed (and other acidic residues are close); however, this difference is quite big. The distance between the oxygen atoms of E238 and D347 of the opposite side of the acidic pocket is 6.5Å in the closed, 3.4Å in the open and 3.8Å in the desensitized conformation. In the short simulations that were conducted for the pKa calculation in the closed state,

calcium ions were placed at the start of the simulation at positions of the AcP and the palm, according to the crystal structure obtained in the presence of divalent ions (Yoder et al., PMID: 30157194) and our recent study (Molton et al., PMID: 38896086). Not only Glu238, but also Glu97 exhibits a pKa value close to ≈ 0.4 . We observed that, in most frames extracted from the short MD simulations of the closed state, a calcium ion simultaneously interacts with both residues at approximately 2.7 Å distance, while the carboxylate side chains of the two glutamates remain 4 to 5 Å apart. Under these conditions, each attractive Coulomb interaction between the calcium and a carboxylate group contributes approximately -3.157 kcal/mol, while the repulsive interaction between the two acidic residues contributes about $+0.947$ kcal/mol. Assuming additive contributions, the net electrostatic interaction energy in this region is estimated at -5.367 kcal/mol. For simulations conducted at 310K, this energy induces a pKa shift of ≈ -3.78 , which likely explains the low pKa values found for these two residues. (We don't discuss this in the manuscript, because we think that there is already a lot of information present).

9. Has ASIC1a pKa data been measured experimentally? If no experimental data exist, how reliable is the computational model?

Response: To our knowledge there exists no experimental confirmation of pKa values in ASICs. Since with the PB approach, we identify many residues that are relevant for ASIC1a function, we consider that it provides useful information. However, we cannot conclude about its reliability. Since the aim of this study was not to test the reliability of the prediction, we did not functionally test residues whose probability of being a pH sensor was low according to the pKa calculations. Since we tested functionally only residues with high priority, our data set cannot be used to estimate the quality of the prediction. The biggest problem in the context of our study could be that with the pKa prediction we might miss important pH sensors. In the revised manuscript we provide a table listing for all titratable, non-cytoplasmic residues, the effect of their mutation on pH dependence if measured in our or in other studies (Table 1 and Supplementary Table 1). We highlight those residues that have not been measured yet and interpret their potential role as pH sensor based on the structural information and pKa calculation (see Supp. Table 1).

10. Lines 212-265: Residues H73, H78, H173, E219, and D409 were previously classified as part of the palm domain but are now referred to as part of the wrist domain. Please clarify this apparent discrepancy.

Response: We consider the wrist as being part of the palm. In the revised version this is better explained. In contrast to what is stated by the reviewer, we classify E219 and D409 throughout the manuscript as part of the acidic pocket. However, there is clearly a problem with placing H173 (which we consider as part of the palm) together with the wrist residues. Also, presenting the H173/E219 mutants in Figure 3 was confusing, we agree entirely. In the revised manuscript, H173 is presented in the supplementary Fig. 4. The H173/E219 double mutants and E219 data are presented in Fig. 5 which deals with the acidic pocket. We have adapted the text accordingly (lines 427-443).

11. Lines 302-303: "K374 shares a proton with both E375 and E413". Does this mean delocalized protonation, a dynamic equilibrium, or direct H-bonding?

Response: We discuss now this aspect in more detail, on lines 333-341.

12. Lines 312-316: The E277 mutations suggest that its titratability affects activation, but pKa values indicate it remains negatively charged in all states. How does a permanently charged residue influence pH gating?

Response: On these lines, we indicate that either the titratability or the negative charge of E277 is important for activation. Based on the analysis of this residue, we did not select it as a "confirmed pH sensor". We conclude that this residue is important for the pH dependence but is likely not a pH sensor. In the revised manuscript the decision tree for the classification is better described. Indeed, there are also some residues whose computational analysis contradicts the functional analysis. This is discussed in the discussion on lines 597-601.

13. Lines 473-474: H73 may drive or stabilize activation? Please, clarify.

Response: The Fraction of protonated His 73 ($f(\text{prot})$) is 0.01 in the closed structure at pH7.4, 0.20 in the closed structure at pH6.0 and 0.22 in the open structure at pH6.0. Since the change in protonation occurs when the pH changes to pH6.0 and persists when the channel is in the open conformation, our interpretation is that protonation of H73 both drives the opening and stabilizes the open state. In the initial manuscript we indicate the calculation of $f(\text{prot})$ in the results and the interpretation on whether a protonation change drives and/or stabilizes a transition in the discussion. In the revised manuscript we have clarified the text in the discussion part (lines 575-601).

14. The Discussion section is very extensive and jumps between computational and experimental findings without always making clear transitions. Please, reorganize findings into clearer parts (e.g., computational predictions, mutagenesis findings, interactions and structural insights).

Response: To make the discussion clearer and have a more logical flow, we have reorganized it completely.

15. The Discussion section refers to "strong" or "small" effects without quantifying these findings. Please, quantify findings where possible to support qualitative descriptions.

Response: We use the decision tree shown in supplementary Fig. 5 to classify residues as putative pH sensors or confirmed pH sensors. In the revised manuscript we have removed the terms "strong" and "small" with regard to the findings of the present study. Among the putative pH sensors, there are some, whose mutation has very small (but significant) effects, and others with "stronger" effects. We consider as "functionally important" residues those, whose conservative mutation induces a $\Delta\text{pH}_{50} \geq 0.2$ or $\Delta\text{pH}_{50} \geq 0.15$. This is now clearly indicated in the text (lines 541-547).

16. Lines 605-606: Protonation states are influenced by pKa, pH, and the local environment (e.g., nearby charged residues, membrane polarity, hydration, etc.), not just pKa and pH alone. Please, clarify the local microenvironmental effects.

Response: We have added a sentence listing the factors on which the pKa depends (line 693-694).

17. Lines 624-630: Membrane interfaces do not behave as uniform dielectrics - they have graded dielectric environments (e.g., lipid headgroups vs. hydrophobic core). The presence of charged lipids and embedded proteins can further alter the electrostatic landscape.

Response: As stated in the Methods, we made use of the capacity of the PBEQ to take, at least partially, the complexity of the membrane into account. For each individual structural image, we calculated the thickness of the membrane, represented by the

position of the phosphorous atoms of both leaflets, and the position of the middle of the membrane in the box as well, represented by the “Z-coordinates” of the lipid methyl groups. This allowed us to attribute specific dielectric values for either the hydrophobic core or the hydrophilic headgroups. However, we agree with the reviewer that, despite these efforts, the accuracy is limited, since it implies that the membrane is viewed as a homogenous ensemble made of parallel layers with discontinuous dielectric constants. This simplistic view of the membrane is inherent to, we think, all currently available pKa solvers. This was already described in the method section and can be found in the present section at lines 820-836.

18. Lines 649-650: It should explicitly state that the Boltzmann distribution accounts for mobile ion distributions in an electrolyte solution (not within the protein itself).

Response: The lines 649-650 (now 736-737) tell that the PB equation is obtained via merging the Boltzmann distribution into the Poisson equation. This statement is thus not related to the mobile ions. However, regarding the mobile ions, we hope that the answer to question 5 helps clarifying the point that, given that the PBEQ takes a protein structure per default, mobile ions, except those nearby ions explicitly described in the PBEQ submitted structure as explained in the main text, can only be considered implicitly.

19. Lines 653-655: The linearized Poisson-Boltzmann equation assumes weak electrostatic potentials, but this approximation breaks down for highly charged surfaces (such as the interior of an ion channel). You should mention that the linearization is valid only when the electrostatic potential is low (e.g., ≤ 25 mV at room temperature).

Response: This linearization may cause a deviation. We added a method paragraph dedicated to the limitation of the PBEQ (lines 759-789) and of our approach.

20. Lines 661-664: Implicit solvation models do not always accurately capture local hydration effects, which can cause errors in pKa calculations. The Born solvation model or explicit solvent corrections are sometimes needed for accuracy.

Response: The use of implicit solvent is indeed a limitation. Yet, we think that our ensemble-averaged PBEQ strategy leverages partially the issue. This is now discussed in the paragraph entitled “limitations of the pKa calculations” mentioned above.

21. Lines 690-691: How does the implicit treatment compare to fully explicit solvent approaches? Is there any benchmarking against experimental data to validate the pKa shifts obtained?

Response: Explicit solvent approaches such as FEP, TI, lambda-based cpHMD or QM/MM certainly address this question more rigorously. This is mentioned now in the ‘limitation of the pKa calculation’ paragraph. On the other hand, to our knowledge, there are also constant pH MD simulations that rely on implicit solvent for steps involved in protonation/deprotonation decision, even if the rest of the algorithm uses explicit solvent. In respect to the specific question of the solvent treatment, these constant pH MD are thus similar to our approach.

22. The study considers ions within 3 Å (monovalent) and 6 Å (divalent) of the protein explicitly while treating the rest implicitly. The chosen distance thresholds seem somewhat arbitrary. Are they based on systematic analysis, or were they optimized to fit experimental pKa shifts?

Response: We used a simple yet practical approach to estimate the distance beyond which the influence of monovalent ions can be considered negligible. A more detailed explanation of this method is now provided in the Supplementary materials methods section together with supplementary Table 4 and supplementary Fig. 9. Admittedly, we did not carry out a comparable analysis for divalent ions. Instead, we approximated their threshold distance by simply doubling that of monovalent ions, which does not strictly adhere to the logarithmic nature of electrostatic interactions.

23. The claim that ignoring ions within this range causes a pKa shift of 1–2.5 units is significant, but is this effect uniform across different protein environments? More validation may be needed.

Response: Our method for estimating the distance beyond which monovalent ions can be neglected does not aim to resolve the underlying complexities of ion screening. Rather, it was intended to provide a practical threshold that improves upon the common practice of simply ignoring nearby ions. We fully acknowledge that a comprehensive understanding of the effects of proximal ions would require much more detailed investigation, encompassing membrane headgroup regions, protein cavities, highly charged or, on the contrary, hydrophobic environments, multi-ion correlation or ‘additive’ effects, crowded environments, and possibly other nuanced contexts. However, such an in-depth analysis lies beyond the scope of the present study.

24. Line 737: Realistic neuronal membranes are not composed solely of POPC – they contain cholesterol, PIP2, and other lipid species that can affect ASIC function. As it is known that ASICs interact with specific lipid domains, was any test done to check if this artificial bilayer affects the protein’s stability?

Response: We agree that this question is highly relevant in typical MD simulations aimed at exploring the phase or conformational space. However, in this particular case, we intentionally performed ultra-short simulations to keep the protein close to its initial conformation while allowing limited side-chain rearrangements, water and ion interactions, and other rapid dynamics. Given the short timescale, membrane composition is not expected to significantly influence the results. This issue has also been addressed in greater detail in a recent publication (Chatelain et al., PMID: 38719838).

25. Line 738: The 0.15 M NaCl concentration is physiologically reasonable but does not account for other ion types (e.g., Ca²⁺, K⁺) that may affect ASIC1a function. Moreover, the authors themselves emphasize in the Discussion the important influence of Ca²⁺ on pH sensors.

Response: Most of the modulatory effects of Ca²⁺ on the channel are thought to result from its occupancy and release at approximate binding sites identified experimentally by the Gouaux lab (e.g., Yoder et al., PMID: 30157194). We placed calcium ions accordingly and conducted intentionally short simulations to preserve the initial coordination environment. Importantly, these simulations are not designed to capture the functional role of calcium; in fact, if such effects were observed on this timescale, they would likely reflect artefactual instability rather than meaningful biological behavior.

26. Line 740: TIP3P is a common but relatively simple water model with known limitations in

describing hydrogen bonding and ion hydration. Why not use TIP4P or TIP5P models, which better represent bulk water properties?

Response: While TIP4P and TIP5P offer improved bulk water properties, we used TIP3P to maintain consistency with the CHARMM force field, which was originally parameterized with this model. This ensures compatibility with protein–solvent and ion–solvent interactions as intended by the force field developers. Moreover, given the focus of our study on relative structural or energetic trends rather than absolute hydration thermodynamics the use of TIP3P remains appropriate.

27. Lines 747-748: Semi-isotropic control assumes membrane fluctuations along the lateral plane, but ASICs undergo large conformational changes. Would anisotropic pressure coupling be more appropriate for a membrane protein undergoing structural transitions?

Response: This is correct. This would indeed represent a limitation, though not an insurmountable one, if the objective were to study conformational transitions. However, as noted in responses to questions 24 and 25, our intention was specifically to avoid inducing large structural rearrangements.

28. Line 753: Linear regression assumes that a single side-chain property has a direct and independent effect on pH50 and pHD50. But residue contributions to pH sensing are often nonlinear and context-dependent, and synergistic effects between size, charge, and flexibility are likely, but linear models do not account for these interactions. Were multiple regression models or machine learning approaches tested? If only single-variable regression was performed, potential multicollinearity effects were ignored.

Response: The limited availability of mutational data precluded the use of more complex statistical approaches. In many cases, there are only five mutants (K388, K384). Computing a two-sample simple t-test in which the pH50 are compared to one single residue feature (size, hydrophobicity, etc.) as we did, leaves us with three degrees of freedom only. If we were to perform a multifactorial ANOVA with, say three features and we oversimplified the features data in two groups each (low, high), we would need eight values to estimate main effects as well as two-way and three-way interactions. In contrast, our simpler approach enabled us to test the association of pH₅₀ shifts with twelve different residue features, at the cost, we acknowledge, of neglecting possible interaction terms.

This limitation becomes even more pronounced in models requiring more data per feature, such as penalized linear regressions (a commonly used ML tool), which would be statistically underpowered given the small number of mutations per residue.

29. Did the authors consider other electrostatic parameters, such as pKa shifts of nearby residues, solvent accessibility, or charge distribution?

Response: Yes, accessing changes due to pKa shifts, materialized by changes in the protonation status, and charge distribution, is the essential purpose of the performed rounds. If the status of residue_A is modified at, say, round 1, the environment felt by nearby residue_B will be different in the round_2, possibly leading us to modify its status at round 2. Notably, solvent accessibility was also accessed, although only indirectly. This last point is detailed in the section “limitation of the pKa calculation”, paragraph ‘hydration and desolvation issues’.

30. Were MD simulations used to check if residue mutations introduce local unfolding or changes in side-chain mobility?

Response: We believe that probing such effects should be carried out through specifically designed studies as we have done previously (see PMID: 31527833, PMID: 38896086, PMID: 36049129, PMID: 32411719 and PMID: 31527833), rather than relying on approximate or general-purpose tools. This is particularly important in the case of subtle or non-obvious mutations, such as Glu → Gln, where the side chains are similar in size and their charge states may only differ depending on protonation, making predictions highly context dependent. Such a study is beyond the scope of this work.

31. Did the authors compare results with ASIC1a from other organisms? If similar correlations exist across species, this would strengthen their conclusions.

Response: In response to a comment of reviewer 1 we provide now a table with previous studies investigating the residues that we report here. Several of these studies were done on rodent ASIC1a and showed consistent results. We did however not go beyond this analysis in the manuscript.

32. Line 766: Please clarify how 5' and 3' non-translated sequences of β -globin improve the protein stability?

Response: This should be RNA stability; it has been corrected.

33. Figures 2–6: Clearly indicate the meaning of the dotted lines used in these figures.

Response: We have added in the legends the following sentence: The dotted lines represent the WT mean pH_{50} and pHD_{50} values.

34. Figures 2–6: Specify the number of individual cells used for measurements. While Figure 2 mentions $n = 7–15$, some bars (e.g., WT, K374Q, etc.) appear to have more than 15 data points. Please verify and clarify.

Response: In Fig. 2, the number indicated refers to the figure 2B, not to the other figures. In the revised manuscript, the n for the pH_{50} and pHD_{50} of each mutant is provided in the table listing the pH_{50} and pHD_{50} values (Supplementary Table 2)

35. Figure 5I: In the right panel, the dotted line for pH_{50} corresponds to 6.7 instead of 6.5, which I assume is the WT control pH_{50} value. This differs from the placement of the dotted line in all other figures. Please explain this shift of 0.2 units. If this is an error, it may be necessary to reconsider the impact of residue K246 when compared to a pH_{50} of 6.5.

Response: This line has now been moved to the correct position. We checked all the other figures and corrected the position of the lines where necessary. This does not change the interpretation.

36. Figure 5H: The dotted line for pHD_{50} is placed at 7.1 instead of 7.15, which is used consistently in all other figures. Please clarify this discrepancy.

Response: This has been corrected.

Reviewer #3 (Remarks to the Author): ligand-gated ion channel functions, form and physiology

Bignucillio et al., 2025 review

This comprehensive work by Bignucolo et al., conducts a systematic assessment of titratable residues in ASIC1a, driven by pK_a estimates, to reinforce and extend existing knowledge about ASIC pH sensing hotspots. The approach, experiments and interpretation are well conducted and justified. This work will be a useful resource for those in the field. Below are my suggestions to enhance the utility of this manuscript.

Major

This project appears to be an updated version for Liechti et al., 2010 JBC. The Liechti paper also used a PB approach to classify residues based on pKa (although limited to fewer structures at the time), followed by functional characterization to identify pH sensors and hotspots. Many (if not most) of the positions identified in the present work were first described by Liechti et al. 2010 or Paukert et al., 2008. It is very important the authors clearly state what new positions or insights arise from this study versus confirmation of past work. In lines 536-541, they discuss other papers that undertook systematic assessments of ASIC1a side chains, but didn't mention the Liechti paper in this context. The authors should directly and explicitly compare this work with the prior Liechti paper to highlight for the reader what new positions, or interactions have been identified and how their present approach has improved (or not) upon that of ~ 15 years ago.

Response: In the revised manuscript we provide a table (Table 1) that shows for each residue that was functionally studied here, the results of previous studies. We discuss what is new and what is just confirmation in the discussion on lines 603-623.

The comprehensive and methodical nature of this data is both a strength and weakness. There is so much here it is difficult to navigate through. Therefore, the authors should pay particular attention to their summary figures. For example, Figure 2I contains a great deal of information but it is too cluttered. The authors use color for both domains, such as blue for thumb and yellow for palm, as well as for the effect size of mutation. The mutations are shown as spheres and the spheres are colored but it is still too confusing. I suggest the authors use color only for the sphere mutations, leaving the cartoon backbone in grey. This should allow the mutations to stand out much better.

Response: This change has been made in the revised manuscript

Also, Figure 7 contains attempts to summarize too many things to actually be useful. Consider removing the boxes around the descriptions of "putative pH sensors" while leaving those around "confirmed". This should reduce clutter and increase readability.

Response: We agree that this figure is very busy. In the revised manuscript we removed all the residues which are not "confirmed pH sensors" if their conservative mutation induced a pH50 shift of < 0.2. This change makes the figure better readable.

In the legend of Figure 7, the authors refer to a position as either "driving" or "stabilizing" the transition based on "changes in protonation fraction". Table 2 further refers to protonation fractions as being in closed, C->O, O, C-> D or End SSD. How do the authors distinguish pKa's during a transition as opposed to endpoints?

Response: We have pKas for the three conformations C, O and D and we assume typical pH conditions, thus 7.4 for the closed channel, a pH6.0 to induce activation and pH6.8 to induce SSD. Based on pKa and pH we can calculate the percentage of protonation (that we term protonation fraction). Therefore, we can determine them at the start of the transition (i.e. for activation, we indicate this moment as "Start activation, C→O", thus corresponding to the moment when the channel is still in the closed conformation, but the pH has changed from 7.4 to 6.0, i.e. closed structure, pH6.0, short C-pH6.0), and at the endpoint of the transition, when the channel has reached the open conformation (O-pH6.0). From the pKa and the pH we calculate the protonation fraction $f(\text{prot})$. We calculated the difference of $f(\text{prot}, \text{C-pH6.0}) - f(\text{prot}, \text{C-pH7.4})$, and $f(\text{prot}, \text{O-pH6.0}) - f(\text{prot}, \text{C-pH7.4})$. If these differences are >0.15 (i.e. when the $f(\text{prot})$ changes at the moment of pH change), we consider the

residue in the first case as a "driver" and in the second case as a "stabilizer". We have reworded the description of this analysis in the discussion (lines 575-592).

The methods mention this implementation of the PB approximation considers the extent to which neighboring residues are also protonated (lines 673-675). This is such an important point that the authors should also mention in the Results or Introduction somewhere. The reader shouldn't have to go digging to answer this fundamental question!

Response: This aspect is mentioned in the introduction (lines 83-101), which refers to the Methods for more detailed information.

The authors should consider a scatter plot of delta pKa between states versus delta pH50 or pH50 to determine how well the PB prediction predicts sites. They should also consider a scatter plot of delta pH50 versus delta pH 50 to better indicate which side chains influence activation or SSD or both.

Response: Regarding the delta-pKa vs. delta pH50 plots, this is an interesting proposition. The problem here is that we mutated only residues that were predicted with a high probability as being pH sensors. Since the aim of the study was not to test the quality of the prediction, we have not mutated residues that were not predicted to be pH sensors. Therefore, we don't have the data to perform a meaningful analysis of this type. Regarding the possible correlation of mutant effects on activation and SSD pH dependence, we provide as Supplementary Fig. 6, plots of ΔpH50 (of SSD) as a function of ΔpH50 (of activation), organized per domain. This shows a correlation in the palm, acidic pocket and the thumb.

Minor

Introduction, Line 54 should include a "the" before "wrist"

Response: This has been changed.

Introduction, the PB approach been used in prior work such as Liechti et al. to study protonation. This paper, or others, should be cited for precedent.

Response: We have adapted the introduction and cite the Liechti study (Line 69).

Line 244, double mutant cycle analysis is prone to false positives and negatives (see PMID 25311637)

Response: When discussing this approach, we mention that there can also be problems with this approach (lines 289-290). Consequently, we interpret the results of the mutant cycle analysis conservatively.

Line 434, "analysis confirmed six residues as true pH sensors". What is a "true" pH sensor? And how could other residues "contribute critically to pH dependence" (next line) but not be "true" sensors? Some clarification, and also identification of these "true" sensors here is needed.

Response: In the revised manuscript we replace "true" by "confirmed" ("confirmed pH sensor", the term that we use throughout the manuscript for residues for which the detailed functional analysis indicates that they are pH sensors. Regarding the second category of residues, which "contribute substantially to pH dependence": We consider as pH sensors for activation (or SSD) residues whose side chain is titratable, and for which our analysis

indicates that titratability is important for their contribution to ASIC function. It has been shown in many other studies on ASICs that mutations of residues that are not titratable can affect pH dependence, most likely because the mutation affects steps leading to activation or SSD, but come after the initial pH sensing event. Therefore, it is likely that some titratable residues whose mutation affects the pH dependence, do not affect the pH sensing event per se, but either affect the sensitivity of pH sensors or affect a subsequent step. We now clearly provide criteria for this classification. This aspect is discussed on lines 529-553.

Figure 3I, H173/E219 and D409 are not the “wrist” domain as the figure title suggests.

Response: We have reorganized the figures and moved these parts of the figure to Fig. 5, which deals with the acidic pocket, and the data on H173 mutations to supplementary Fig. 4.

Line 438 “provides a robust framework for identifying pH-sensing mechanisms in other ion channels”. Some other channels should be mentioned.

Response: Since we have reorganized the Discussion, this statement has disappeared. We would however think of other pH-sensitive channels, such as PAC or OTOP.

Dear reviewers,

We thank all reviewers for their careful evaluation of our manuscript and for their insightful comments that helped us improve the manuscript.

Below we indicate the point-to-point responses.

Sincerely,

Stephan Kellenberger

Reviewers' comments:

Reviewer #1 (Remarks to the Author):

All my comments are addressed carefully and correctly.

Reviewer #2 (Remarks to the Author):

The authors have addressed the majority of my previous comments and questions. However, a few critical points remain that still require attention:

1. Please clarify the changes made to Fig. 1D. It appears to differ from the previous version. Please verify its accuracy and include confidence intervals in the figure.

Response: Thank you for bringing this up. For some reason, the SD was not shown on Fig. 1D, and there was a different order of the states, as also remarked by reviewer #3. We have now re-placed the error bars in Fig. 1D. We have also changed in Fig. 1C-1I the order, in which the symbols for the conformational states appear, as (from left to right) closed - open - desensitized. Please see the revised figure at the end of this file.

The values are correct. As mentioned in the initial revision, the controls during the revision identified some errors in pKa values that had been corrected in the revision (concerning only a very small number of residues). This makes that some pKa values are different between the initial submission and revision 1.

2. Line 32: I suggest specifying that the desensitization referred to is "steady-state desensitization" for clarity.

Response: This change in the abstract has been made.

3. Line 44: "ASIC3a, -3b and -3c" applies specifically to human ASIC3, not to ASIC3 isoforms across all mammals. Please revise this accordingly to avoid overgeneralization.

Response: This has been reformulated: Four genes encode eight different ASIC subunits in mammals, ASIC1a and -1b⁴⁻⁷, ASIC2a and -2b^{5,8}, ASIC3 (in rodents; ASIC3a, -3b and 3c in humans)⁹⁻¹¹ and ASIC4¹²⁻¹⁴.

4. Line 676: The phrase "determinators of pH dependence" should be replaced with more standard scientific language, such as "key determinants of pH sensitivity".

Response: This change has been made: Based on this and previous studies we conclude that titratable residues of the AcP, the palm and the wrist are the key determinants of pH

dependence,...

5. Conclusion: I recommend closing the paragraph with a forward-looking statement that highlights how these insights could guide future investigations into other ASIC isoforms, or that places the findings in a broader physiological or pharmacological context, for example, by mentioning potential implications for ASIC-related diseases or therapeutic development.

Response: We have added a sentence describing the potential application of the information provided by our study: Our in-depth analysis of ASIC1a pH sensing adds new elements towards a molecular description of the ASIC activation mechanism, and highlights critical sites within the channel that could serve as targets for new drugs.

Reviewer #3 (Remarks to the Author):

The authors have done substantial revisions to improve clarity and presentation. While this is still a huge amount of work, and difficult to parse through, I do not have any further suggestions to improve. Here are some minor thoughts:

In reading the response to R1, the authors state they have an $n = 15$ or 8 . But they describe this as at each amino acid they measure protonation at 3 subunits and 5 time points, thus giving $n = 15$. I disagree with this interpretation. N is commonly treated as independent replicates. Counting the same amino acid at multiple points in time is not independent. The assumption that each subunit is independent is also shaky but less concerning.

Response: We thank the reviewer for raising this important statistical question. This touches on a fundamental distinction between 'object variability' versus 'measurement error' in the context of dynamic biological systems. We think that there is no pseudoreplication in this specific case. The observed pK_a variability reflects the sensitivity of the PBEQ solver to the local atomic environment surrounding each ionizable group. As detailed in the manuscript and our previous responses, this local environment captures the effects of hydration shell dynamics, ion movements, and subtle conformational changes that occur during the MD trajectory and are encoded in the protein coordinates. Regarding the 5 time points, each ionizable residue was evaluated five times using snapshots separated by 2 ns. Critically, if we had used a static crystal structure five times, the PBEQ solver would yield identical results (within numerical precision). The variability we observe arises because the ionizable group genuinely experiences different electrostatic environments due to hydrogen bond rearrangements, side chain rotamer fluctuations, and water/ion repositioning, even within this short timeframe. This is not pseudoreplication because we are not claiming temporal independence to study kinetics. If we did, the time interval of 2 ns would be insufficient. Rather, we are sampling the conformational ensemble to characterize the electrostatic plasticity of each ionizable site. Each 2 ns snapshot represents a legitimate member of the conformational distribution that the protein explores under physiological conditions, in the limits of the force field

accuracy. The resulting pKa variability reflects the intrinsic thermodynamic fluctuations of the local electrostatic environment.

Regarding the 3 subunits, while subunits are crystallographically identical, they rapidly diverge during MD simulation, reflecting the asymmetry that proteins exhibit in solution.

The standard deviation across our 15 measurements thus quantifies the conformational pKa variability, a parameter that static calculations would miss. We acknowledge that a single trajectory of 10 ns simulations provides limited conformational sampling, due to computational constraints. However, the observed variability captures genuine electrostatic fluctuations that occur on the timescales sampled, providing insights into the dynamic nature of protein electrostatics that complement static structure-based predictions.

Intro, line 80 uses "based" twice in the same sentence.

Response: This sentence has been reformulated to avoid the repetition of the word "based": Ensemble-based pKa calculations from structural models of the closed, open and desensitized conformations of ASIC1a were used to predict potential pH-sensing residues.

Figure 1, in panel D, the symbols within a column are in the following order: closed, open and desensitized. In the other panels, the order is open, desensitized, closed. Why the difference?

Response: This was a mistake. Reviewer #2 had also seen that the error bars were missing in Fig. 1D. Since it seems more logical to have the order closed - open - desensitized, we have applied this order to all pKa panels of Fig. 1 (Please see the updated Fig. 1 on the next page).

Figure 5, panels B – D, please double check the use of K211D versus K211E in these panels. The ms text, figure legend and figure labels might be conflicting at times.

Response: The complication occurred since as single mutant, K211E was measured, while in the double mutants, K211D was used. We consider that the use of these different acidic residues does not affect the conclusion, however, we made some errors in the description. We have corrected the text (line 401) and the figure legend accordingly (lines 1363-1364).

Figure 1

Rebuttal letter for **Molecular mechanisms and hotspots of pH sensing in ASIC1a revealed by computational and functional analysis**

by Olivier Bignucolo, Ophélie Molton, Ivan Gautschi and Stephan Kellenberger

This revision addresses formal changes in the manuscript. All comments by reviewers have already been addressed in the previous revision.